EMBO
Molecular Medicine

# Lanatoside C activates the E3 ligase STUB1 to inhibit FOXP3 transcriptional activity and promote antitumor immunity

Qian Zhou [1,6✉], Tong Yang [1,6], Xixi Yu[1,6], Bo Li[2,6], Jin Liu[1,6], Yongxin Mao[2], Rongxiang Guo[1], Zhuo Feng [1], Li Zhou[2], Guandi Zeng [1], Nan Li[1], Jinxia Liang[1], Lu Liu[1], Pengju Feng[3], Hong-Bing Shu [4✉] & Liang Chen [1,5✉]

## Abstract

**Regulatory T cells (Tregs) play critical roles in inhibiting antitumor immunity, which is dependent on FOXP3-mediated transcriptional activity. However, no Treg-specific therapeutics has been approved for clinical use. We performed a high-throughput screen of FDA-approved drugs for potential inhibitors of FOXP3 transcriptional activity. These efforts identified Lanatoside C (Lac), which potently inhibits FOXP3 activity by causing degradation of RUNX1, a FOXP3-associated component required for its transcriptional activity. Lac directly binds the E3 ligase STUB1, leading to increased poly-ubiquitination and proteasomal degradation of RUNX1. Lac inhibits Tregs activity and promotes antitumor immunity in a mouse primary lung cancer model. In addition, Lac synergizes with PD-1 inhibitor to shrink lung cancers driven by mutant KRAS in a mouse model. Our findings suggested that the FDA-approved Lac is a Tregs inhibitor and serves as a candidate drug for cancer patients by its own or in combination with existing therapeutics such as PD-1 inhibitors.**

**Keywords** Immunotherapy; Tumor Microenvironment; Regulatory T Cells; PD-1 Inhibitor
**Subject Categories** Cancer; Immunology

## Introduction

Immunotherapy has changed the paradigm of cancer therapy. Exploiting new knowledge based on immune evasion mechanisms, immunotherapies, as exemplified by anti-PD-1 therapy, normalize the antitumoral immune response to treat certain cancer (Sanmamed and Chen, 2018). Unfortunately, majority of cancer patients fail to benefit from these efforts. There is an urgent need to develop new strategy for cancer treatment.

The immunosuppressive tumor microenvironment (TME) represents the major hurdle for successful oncological immunotherapies. TME normalization is a clinically tested treatment strategy to stimulate antitumor immunity and enhance immunotherapy efficacy. Among the TME cell components, infiltration of regulatory T cells (Tregs) is inversely proportional to antitumor activity and the survival of cancer patients. Tregs exert potent immunosuppressive function through various mechanisms, including CTLA4-mediated suppression of antigen-presenting cells (APCs), IL-2 consumption through CD25, conversion of ATP into adenosine, production of inhibitory cytokines (IL10 and TGF-β), and cytolysis of effector T cells (Josefowicz et al, 2012). Therefore, Treg is one of the most intensively studied target for immunotherapy. Antibodies have been tested to deplete Tregs, including those against CD25 (Shimizu et al, 1999), CTLA4 (Selby et al, 2013; Simpson et al, 2013), OX-40 (Bulliard et al, 2014), GITR (Bulliard et al, 2013) and CCR4 (Sugiyama et al, 2013). Unfortunately, no single antibody is capable of distinguishing Tregs from other cell populations. Tregs are also reported to be sensitive to low dose of cyclophosphamide (Ercolini et al, 2005; Lutsiak et al, 2005). However, cyclophosphamide is highly toxic through the alkylation of cellular DNA.

FOXP3 is the lineage-defining transcription factor for Tregs. Analysis of mice with disruption of the FOXP3 protein expression through GFP knocked into its locus revealed that Foxp3 was not necessary for the survival of Tregs precursors, but was essential for Tregs' suppressor function (Gavin et al, 2007). FOXP3 was found to amplify and stabilize molecular features of Tregs precursor cells, and thus important for function and maintenance of Tregs (Gavin et al, 2007). This conclusion was further strengthened through

[1]State Key Laboratory of Bioactive Molecules and Druggability Assessment, Guangdong Basic Research Center of Excellence for Natural Bioactive Molecules and Discovery of Innovative Drugs, College of Life Science and Technology, Jinan University, 510006 Guangzhou, China. [2]MOE Key Laboratory of Glucolipid Metabolic Diseases, Guangdong Metabolic Diseases Research Center of Integrated Chinese and Western Medicine, College of Chinese Medicine Research, Guangdong Pharmaceutical University, 510006 Guangzhou, China. [3]Department of Chemistry, College of Chemistry and Materials Science, Jinan University Guangzhou, 510632 Guangzhou, China. [4]Department of Infectious Diseases, Medical Research Institute, Zhongnan Hospital of Wuhan University, Wuhan University, 430073 Wuhan, China. [5]MOE Key Laboratory of Tumor Molecular Biology and Key Laboratory of Functional Protein Research of Guangdong Higher Education Institutes, Institute of Life and Health Engineering, College of Life Science and Technology, Jinan University, 510632 Guangzhou, China. [6]These authors contributed equally: Qian Zhou, Tong Yang, Xixi Yu, Bo Li, Jin Liu. ✉E-mail: zhouqian@jnu.edu.cn; shuh@whu.edu.cn; chenliang@jnu.edu.cn

analysis of the transcriptome of FOXP3-expressing and non-expressing cells generated in vivo and in vitro (Gavin et al, 2007; Hill et al, 2007). Intriguingly, there seems to be a threshold of FOXP3 protein level to be functional, since Tregs fail to suppress spontaneous autoimmunity in transgenic mice harboring FOXP3 gene with manipulated 3′ untranslated region (UTR) for translating about tenfold decrease in Foxp3 protein (Wan and Flavell, 2007). It is, therefore, tempting to inhibit Tregs through targeting FOXP3.

STUB1 (STIP1 homology and U-box-containing protein 1) was identified as the first chaperone-associated E3 ligase that targets Hsp90 (partially folded) and Hsp70 (misfolded) clients for proteasomal degradation (Connell et al, 2001). STUB1 encodes a protein of 303 amino acids, featuring an N-terminal tetratricopeptide repeat (TPR) domain, a middle coiled-coil domain (CC), and a C-terminal U-box domain (UBOX) (Paul and Ghosh, 2014). STUB1 plays an important role in inflammatory response, autoimmunity, antiviral, and anti-tumor immune responses (Hu and Sun, 2016).

Ono et al reported that FOXP3 recruits RUNX1 to form a protein complex, which is necessary for its transcriptional function of Tregs-associated molecules and their suppressive activity (Ono et al, 2007). Here, we reported high-throughput screening for the identification of inhibitors capable of breaking down the FOXP3/RUNX1 complex. We found that Lanatoside C (Lac) potently disrupted the FOXP3/RUNX1 complex and inhibited FOXP3 transcriptional activity and Tregs function. Mechanistically, Lac directly bound an E3 ligase, STUB1, and enhanced ubiquitination and subsequent proteasomal degradation of RUNX1, thereby inhibiting FOXP3 transcription activity. Lac potently rewired TME through inhibiting Tregs in vivo. We also showed Lac synergized with PD-1 inhibitors to shrink lung cancers driven by mutant KRAS.

## Results

### High-throughput screening of drugs capable of disrupting FOXP3-RUNX1 complex

It has been reported that Tregs' master transcription factor, FOXP3, recruits RUNX1 to form a complex and thereby controls the expression of a series of target genes important for the function of Tregs (Ono et al, 2007). In order to identify small molecular drugs capable of blocking the interaction between FOXP3 and RUNX1, we generated a stable 293T cell clone stably co-expressing N- and C-terminal half of luciferase fused FOXP3 and RUNX1 (designated Nlu-FOXP3-Flag and RUNX1-Clu-Myc respectively, designated 293T-NF + RC for the engineered cell clone), such that luciferase activity directly reflects the strength of the interaction between FOXP3 and RUNX1 in bimolecular luciferase complementary assay (Bais et al, 2023) (Fig. 1A). Consistently, we detected robust luciferase activity in 293T-NF + RC, but not in 293 T co-expressing FOXP3-Nluc and CD27-Clu-Myc (NF + CC, served as a control) (Appendix Fig. S1A,B), arguing that 293T-NF + RC is a convenient platform for evaluating the ability of a drug to inhibit the interaction between FOXP3 and RUNX1. We then screened library of FDA-approved drugs and identified 48 chemicals that inhibited luciferase activity by more than twofold (Fig. 1B). Among these 48 candidates, 6 drugs were verified to dramatically inhibit the luciferase activity of 293T-NF + RC without significant cell cytotoxicity (Appendix Fig. S1C,D). Interestingly, we found

Lanatoside C (Lac, P20-H8) (Appendix Fig. S1E), potently inhibited the luciferase activity of 293T-NF + RC in a dose- and time-dependent manner (Appendix Fig. S1F,G), strongly suggesting Lac as an inhibitor against FOXP3/RUNX1 complex.

Ono et al reported that FOXP3 and RUNX1 formed a complex to transcribe target genes important for the suppressive function of Tregs (Ono et al, 2007). We therefore went to test the impact of Lac on the transcription of those genes. We purified Tregs from lymph node through magnetic-activated cell sorting (MACS) (Fig. EV1A) and found that Lac potently inhibited the transcription of Cd25, Ctla4, and Il10, 3 typical substrates of FOXP3 (Ono et al, 2007), but not IFNγ (Fig. 1C). Moreover, we found that incubation of Tregs with Lac led to significant reduction of CD25[+] and CTLA4[+] population (Figs. 1D,E and EV1B–E). Furthermore, Lac treatment significantly inhibited Perforin (PER) and IL10 expression in Tregs (Figs. 1F–H and EV1F–I). Lac also suppressed the population of CD25[+] and CTLA4[+] in in vitro differentiated Tregs (Fig. EV1J,K). We went further to test the impact of Lac on the viability of Tregs. Interestingly, we found that Lac significantly inhibited proliferation (Figs. 1I,J and EV1L,M), but not apoptosis of Tregs, in response to αCD3/28 stimulation (Fig. EV1N,O).

Tregs demonstrate plastic differentiation under certain circumstances to exhibit features of helper T (Th) cells, as characterized by secretion of Th-related cytokines and expression of specific transcription factors, while retaining the expression of FOXP3 (Qiu et al, 2020). To check the impact of Lac on the differentiation of Tregs, we purified Tregs and treated them with Lac. Results showed that Lac suppressed FOXP3 expression (Appendix Fig. S2A,B), but did not significantly affect the expression of Th1 and Th2 related cytokines by Tregs (Appendix Fig. S2C,D).

Taken together, we identified Lac as an inhibitor of FOXP3 transcription activity through inhibiting formation of FOXP3/RUNX1 complex and that Lac potently inhibited differentiation and proliferation of Tregs.

### Lac alleviates the inhibitory effect of Tregs on CTLs

Tregs present a formidable hurdle in tumor immunotherapy by inhibiting the function of CD8[+] cytotoxicity T cells (CTLs). We went on to test the impact of Lac treatment on the inhibitory function of Tregs on CD8[+] CTLs. Of note, Lac per se did not affect the proliferation of CD8[+] T cells (Fig. EV2A,B) and had a limited impact on the expression of Granzyme B (GRZB) (Fig. EV2C,D) and IFNγ by CTLs (Fig. EV2E,F). Lac was also non-toxic to EG7 cell, the target of OT-I T cell (Fig. EV2G,H). To test the impact of Lac on the killing ability of CTLs, we isolated splenic T cells and stimulated them with OVA257-264 peptide for 3 days following our earlier report (Fan et al, 2020) (Fig. EV2I). We found that Lac treatment dramatically compromised Tregs' ability to inhibit the proliferation of CTLs (Fig. 2A,B). Of note, Lac per se did not affect the killing ability of CTLs on target cells (Fig. EV2J,K). Moreover, Lac abolished Tregs' ability to inhibit the expression of IFNγ and GRZB (Fig. 2C–F) and cytotoxicity (Fig. 2G,H) by CTLs. Collectively, these results suggested that Lac alleviated the inhibitory effect of Tregs on CTLs.

### Lac promotes the degradation of RUNX1 protein

Our above results have shown that Lac potently inhibited the interaction between FOXP3 and RUNX1 as Lac inhibited the

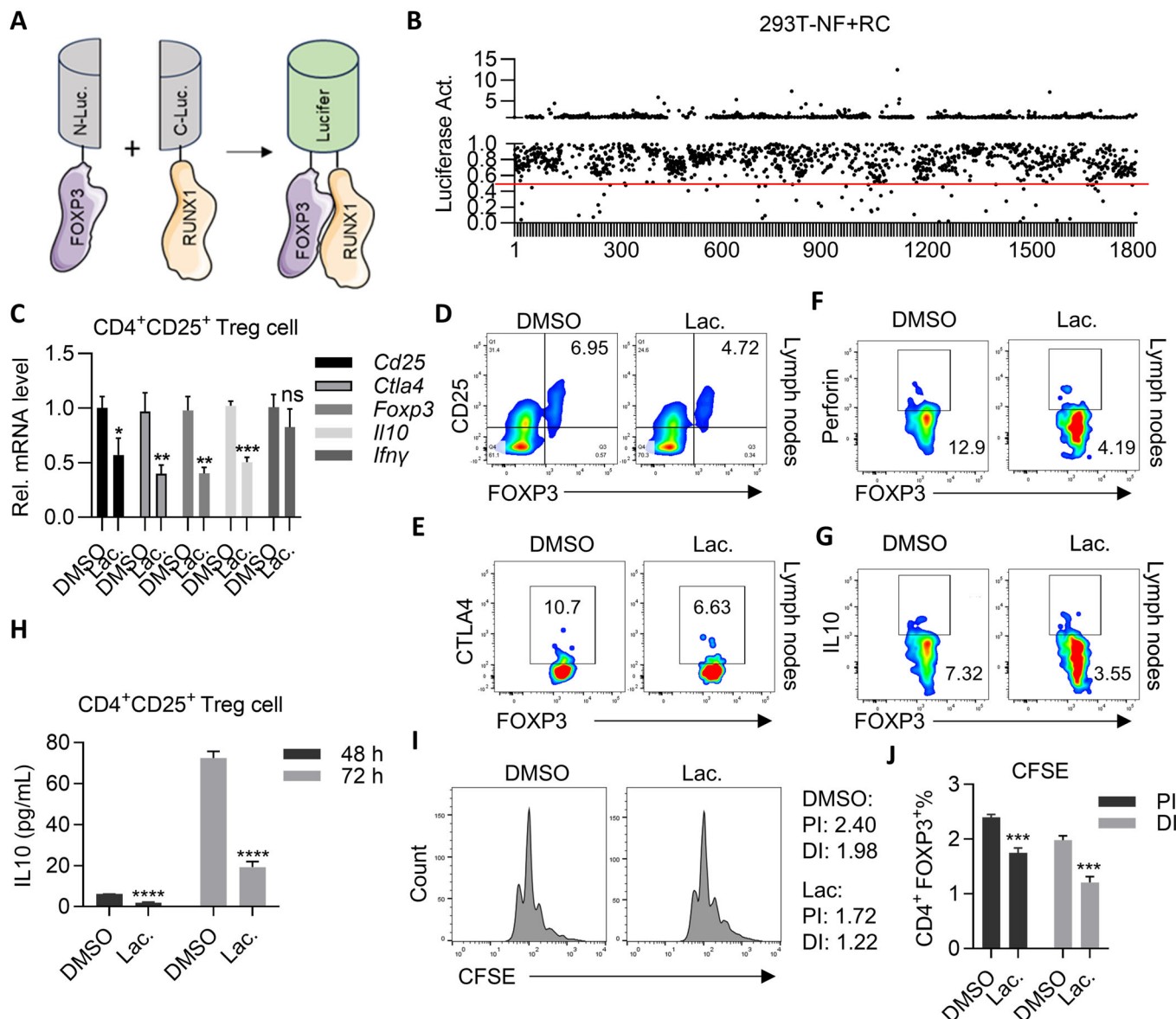

**Figure 1. High-throughput screening of drugs capable of disrupting FOXP3-RUNX1 complex.**

(A) Schematic diagram for high-throughput luciferase complementation screening. (B) Screening of FDA-approved drugs library for compounds capable of inhibiting luciferase activity in 293T-NF + RC. (C) Lac suppresses transcription of *Foxp3, Il10, Cd25, Ifnγ* and *Ctla4* in Tregs. Data are representative of three independent experiments and were analyzed by Student's *t* test. Error bars denote mean ± SD. *P* value: *Cd25*, *$P = 0.0157$; *Ctla4*, **$P = 0.0065$; *Foxp3*, **$P = 0.0021$; *Il10*, ***$P = 0.0001$; *Ifnγ*, $P = 0.1926$. (D–G) Flow cytometry analysis of the impact of Lac on marker protein expression by Tregs. (H) Lac suppresses IL10 expression by Tregs. Tregs ($2 \times 10^5$) were isolated from lymph nodes which were treated with DMSO or Lac for 48 or 72 h before ELISA analysis. Data are representative of three independent experiments. ****$P < 0.0001$ by Student's *t* test. Error bars denote mean ± SD. (I) Lac suppresses the proliferation of Tregs in vitro. CFSE-labeled Tregs in lymph nodes were ($2 \times 10^5$) treated with DMSO or Lac for 48 h. (J) Statistics analysis of (I). Data are representative of three independent experiments and were analyzed by Student's *t* test. Error bars denote mean ± SD. *P* value: PI, ***$P = 0.0004$; DI, ***$P = 0.0006$. Source data are available online for this figure.

luciferase activity of 293T-NF + RC (Appendix Fig. S1F,G). To elucidate the underlying mechanism, we went further to test this inhibitory effect through Co-IP experiments. Our data showed that treatment of 293T-NF + RC with Lac potently inhibited the ability of FOXP3 to precipitate RUNX1 (Fig. 3A). Interestingly, we found that RUNX1 protein level was significantly downregulated in response to Lac treatment (refer to line 4 "RUNX1-Myc" western blot result on whole-cell lysate in Fig. 3A). This was further confirmed in Jurkat cells overexpressing FOXP3 (designated Jurkat-FOXP3, Appendix Fig. S3A,B). Likewise, we found Lac treatment reduced the amount of RUNX1 precipitated by FOXP3-Flag (Dataset EV1 (DMSO group: 10 peptides, Lac group: 2 peptides)). These results suggested that Lac suppressed the interaction between RUNX1 and FOXP3 by degrading RUNX1.

To validate this phenomenon, we generated a stable 293T cell line for overexpressing RUNX1 and treated it with Lac. We found that Lac

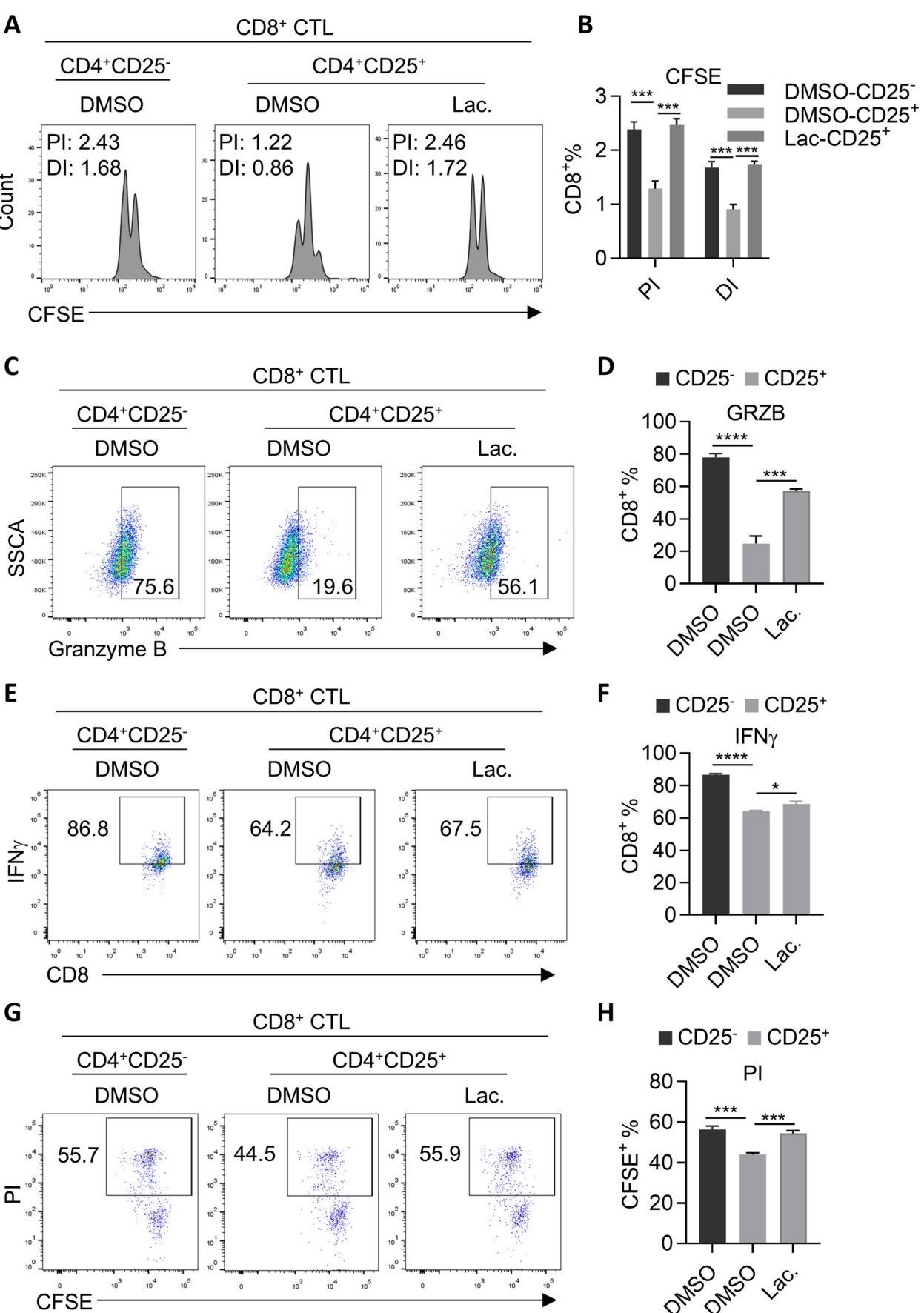

**Figure 2. Lac alleviates the inhibitory impact of Tregs on CTLs.**

(A) Lac alleviates inhibition of CTLs proliferation by Tregs. (B) Statistics analysis of (A). Data are representative of three independent experiments and were analyzed by Student's *t* test. Error bars denote mean ± SD. *P* value: PI (DMSO-CD25⁻ vs DMSO-CD25⁺), ***P = 0.0007; PI (DMSO-CD25⁺ vs Lac-CD25⁺), ***P = 0.0003; DI (DMSO-CD25⁻ vs DMSO-CD25⁺), ***P = 0.0007; DI (DMSO-CD25⁺ vs Lac-CD25⁺), ***P = 0.0002. (C–F) Lac alleviates Tregs-induced suppression of CTLs expression of GRZB and IFNγ. Expression of GRZB (C) and IFNγ (E) were detected by flow cytometry. Statistics analysis of (C, D) and (E, F). Data are representative of three independent experiments and were analyzed by Student's *t* test. Error bars denote mean ± SD. *P* value: GRZB (DMSO-CD25⁻ vs DMSO-CD25⁺), ****P < 0.0001; GRZB (DMSO-CD25⁺ vs Lac-CD25⁺), ***P = 0.0003; IFNγ (DMSO-CD25⁻ vs DMSO-CD25⁺), ****P < 0.0001; IFNγ (DMSO-CD25⁺ vs Lac-CD25⁺), *P = 0.0174. (G) Lac alleviates the inhibition of cytotoxicity of CTLs by Tregs. (H) Statistics analysis of (G). Data are representative of three independent experiments and were analyzed by Student's *t* test. Error bars denote mean ± SD. *P* value: DMSO-CD25⁻ vs DMSO-CD25⁺, ***P = 0.0004; DMSO-CD25⁺ vs Lac-CD25⁺, ***P = 0.0005. Source data are available online for this figure.

potently downregulated RUNX1 protein level (Fig. 3B). Of note, Lac did not change mRNA level of RUNX1 (Appendix Fig. S3C). Consistent with earlier report in T cell (Ono et al, 2007), we found that Jurkat cell expressed high level of RUNX1 (Fig. 3C). Importantly, Lac potently downregulated endogenous RUNX1 protein level (Fig. 3C–E), but not its mRNA level (Appendix Fig. S3D). This effect was also seen on EL4, a mouse T lymphoblast cell line (Fig. 3F; Appendix Fig. S3E). In line with these, cycloheximide (CHX) treatment shortened the half-life of both ectopically expressed and endogenous RUNX1 in these cell lines, which was further shortened by Lac treatment (Fig. 3G,H; Appendix Fig. S3F,G). Our data, therefore, showed that Lac degraded RUNX1 protein, thus decreasing the amount of functional FOXP3-RUNX1 complex.

## STUB1 is required for Lac-induced RUNX1 degradation through proteasome pathway

We went further to elucidate the mechanism underlying Lac's ability to degrade RUNX1. Our thermal shift experiments (Martinez Molina et al, 2013) indicated that Lac did not obviously change the thermal stability of RUNX1 (Fig. EV3A), thus excluding the possibility of Lac binding RUNX1. Protein homeostasis is controlled by lysosomal and/or proteasomal pathways. Interestingly, we found that proteasomal inhibitor MG132, but not lysosomal inhibitor NH4Cl and Chloroquine (CQ), stabilized the protein level of RUNX1 in 293T cell line and Jurkat treated by Lac as revealed by western blot analysis (Fig. 4A,B). Using our 293T-NF + RC cell clone, we also found that MG132, but not NH4Cl or CQ, stabilized luciferase activity in response to Lac treatment, strongly arguing that MG132 stabilized RUNX1 protein level to maintain FOXP3 and RUNX1 interaction and bimolecular Luciferase complementation (Fig. EV3B).

We then went on to identify the E3 ligase responsible for Lac-mediated proteasomal degradation of RUNX1 through Immunoprecipitation-Mass Spectrometry (IP-MS). STUB1 caught our attention among the candidate interactors throughout three biological repeats of IP-MS analysis (Fig. 4C; Table EV1). STUB1 has been reported as an E3 ligase to ubiquitinate RUNX1 for its proteasomal degradation (Shang et al, 2009; Yonezawa et al, 2017). Co-IP experiments confirmed that STUB1 interacted with RUNX1 (Fig. 4D) and enhanced the ubiquitination of RUNX1 (Fig. EV3C). More importantly, Lac significantly enhanced the ability of RUNX1 to precipitate STUB1 (Fig. 4D). Confocal microscopy also revealed that Lac treatment enhanced colocalization between STUB1 and RUNX1 (Fig. 4E), arguing that Lac promoted interaction between RUNX1 and STUB1. We found that STUB1 localized mainly in cytoplasm while detectable in nuclei (Fig. 4E), which was consistent with previous studies (Yonezawa et al, 2017). Strikingly, we found

that Lac treatment significantly increased the distribution of RUNX1 in the cytoplasm (Fig. 4E,F), with around 1% of the cells showing complete absence of RUNX1 in nuclei (Fig. EV3D).

Overexpression of STUB1 led to degradation of RUNX1, which was significantly enhanced by Lac (Fig. 4G). Importantly, our data further showed that Lac enhanced the ubiquitination level of RUNX1 by STUB1 (Fig. 4H). Conversely, knockout of STUB1 with two different sgRNAs (Fig. EV3E) stabilized RUNX1 protein level in 293T in response to Lac treatment (Fig. 4I,J). Moreover, STUB1 knockout stabilized the robust luciferase in response to Lac treatment in 293T-NF + RC cells (Fig. EV3F). Of note, Lac elicited protein degradation mediated by STUB1 showed specificity, as indicated by degradation of RUNX1, but not FOXP3 in 293T-NF + RC in response to treatment with Lac, although FOXP3 was also reported to be a substrate of STUB1(Chen et al, 2013) (Fig. EV3G).

Taken our above data together, Lac enhanced the ability of STUB1 to capture cytoplasmic RUNX1 for ubiquitylation and subsequent proteasomal degradation, leading to reduction of nuclear RUNX1.

## Lac directly binds STUB1 at its TPR domain to enhance its affinity to RUNX1

We went on to investigate the mechanisms underlying Lac's ability to enhance E3 ligase activity of STUB1 on RUNX1. Interestingly, we found that Lac directly bound STUB1 as revealed in the thermal shift assay (Fig. 5A,B). This is further confirmed by our findings that Lac treatment protected ectopically expressed STUB1 from pronase digestion (Fig. 5C,D). STUB1 consists of three structural domains: the N-terminal TPR domain (1–127 aa), the middle CC domain (134–218 aa), and the C-terminal UBOX domain (230–293 aa) (Fig. 5E). We then truncated STUB1 by domains and assayed their thermal shift in response to Lac treatment. We found that Lac significantly enhanced the thermal stability of the N-terminal TPR domain (1-130 aa) of STUB1 (Fig. 5F,G). Most importantly, isothermal titration calorimetry (ITC) assay on purified TPR domain (Fig. EV4A) showed that Lac bound TPR domain of STUB1 with a Ka of 1.30E5 ± 8.54E4 M⁻¹ (Fig. 5H). These results suggested that Lac directly bound TPR domain of STUB1.

Callahan's team recently solved the crystal structure of TPR domain of STUB1 (Callahan et al, 2023). To analyze the detailed interaction between Lac and TPR domain of STUB1, we ran molecular dynamics simulations. We found that Lac molecule lied in the long and narrow groove of the TPR domain (Fig. 5I). Lac molecule was confined in rather fixed location in TPR domain in solution as indicated by limited variation of RMSD value, with an average of only 0.749 during the process of 200 ns dynamics

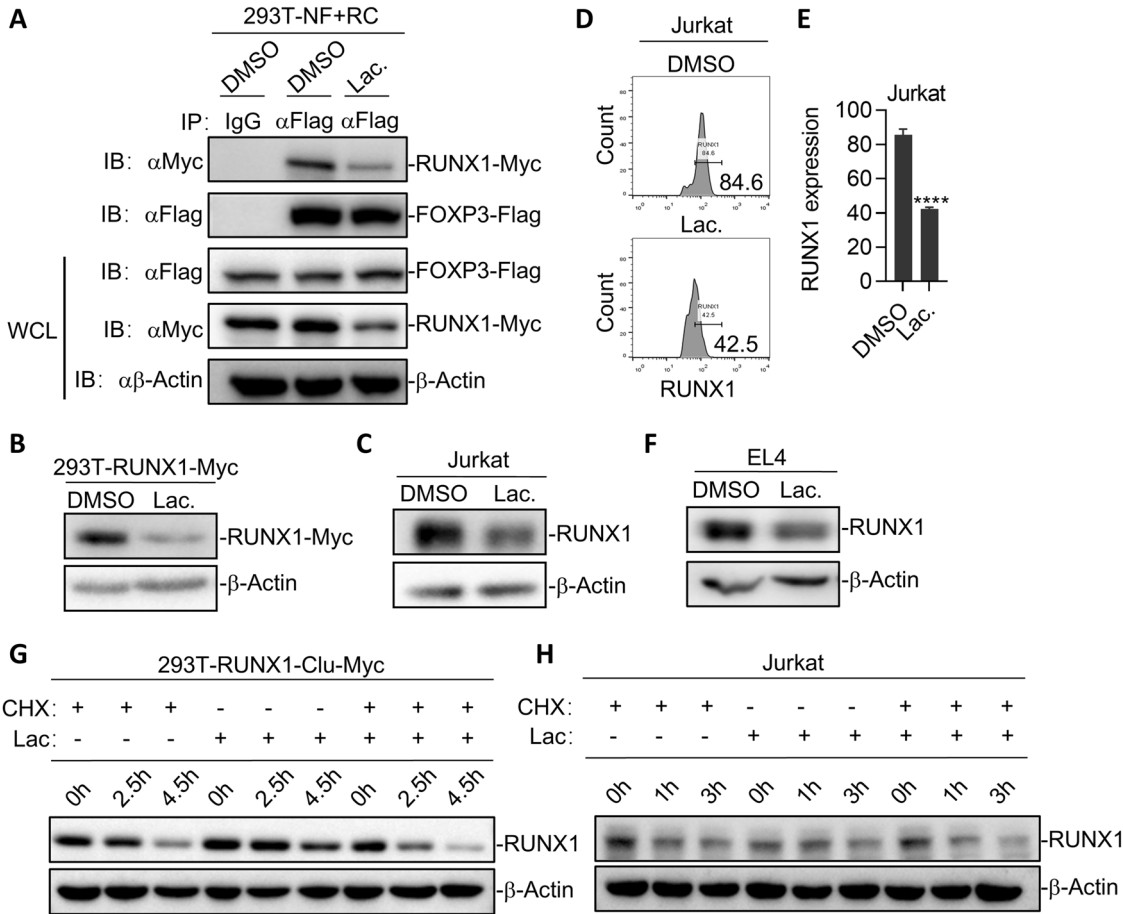

**Figure 3. Lac promotes the degradation of RUNX1 protein.**

(A) Lac suppresses the association between FOXP3 and RUNX1 in 293T-NF + RC. (B–F) Lac promotes the degradation of RUNX1. 293T-RUNX1-Myc (B), Jurkat (C–E), or EL4 (F) cells were treated with DMSO or Lac for 6 h before IB or flow cytometry analysis with indicated antibodies. Statistics analysis of (D, E). Data are representative of three independent experiments and were analyzed by Student's t test. Error bars denote mean ± SD. P value: DMSO vs Lac, ****P < 0.0001. (G, H) Analysis of RUNX1 degradation by cycloheximide (CHX) pulse. 293T-RUNX1-Clu-Myc (G) or Jurkat (H) cells were treated with CHX, Lac or CHX+Lac for the ndicated duration. IB was performed with indicated antibodies. Source data are available online for this figure.

simulation (Fig. 5J). In parallel, the movement of each atom in Lac molecule, as indicated by RMSF value, was low (Fig. EV4B). These results suggested that the above-mentioned binding mode was stable.

Careful study of the entire process of dynamics simulation revealed that the H atom of amino group of Lys72 alkyl chain end formed hydrogen bond with O atom of Lac sugar part (Fig. EV4C). We also observed that the RMSD value of Lac fluctuated to a maximum value of 1.028 at 124.14 ns (Fig. 5J), when amino group of Gln102 formed a hydrogen bond with O atom of glycosidic bond between sterol core and sugar part (Fig. EV4C). Molecular dynamics simulation revealed that Lys72Ala and Gln102Ala TPRs bound Lac at higher free energy than WT TPR domain by 0.65 and 1.70, respectively (Fig. EV4D). In addition, the fluctuation of Lac's motion trajectory increased significantly in these mutant TPRs during 200 ns of simulation (Fig. EV4E), typical of unstable binding. These results indicated that Lys72 and Gln102 played an important role in mediating TPR's binding of Lac. Most importantly, our thermal shift data revealed that K72A or Q102A mutant STUB1 lost their ability to bind Lac (Fig. 5K,L). Moreover, Lac lost its ability to protect these mutants from

digestion by pronase in drug affinity responsive target stability (DARTS) assay (Pai et al, 2015) (Fig. EV4F,G). Notably, K72 and Q102 are conserved among species (Fig. EV4H), suggesting that Lac may be able to bind STUB1 protein among various animal species.

Taken together, our data showed that Lac bound directly to TPR domain of STUB1, with K72 and Q102 playing a critical role.

## Lac rewires tumor microenvironment by inhibiting the function of Tregs

Lac has been prescribed to treat congestive heart failure and cardiac arrhythmia (Kelly, 1990). More importantly, the favorable safety profile of Lac has been well demonstrated in earlier reports (Dresdale et al, 1959; Polis et al, 1971). We then went on to test its in vivo tumor-therapeutic efficacy. To this end, we grew LLC tumors on immunocompetent C57BL/6J host mice (designated B6 mice) and randomized them for treatment with Lac or vehicle when tumors reached a volume of around 60 mm³. We found that Lac significantly inhibited the growth rate of LLC tumor (Fig. 6A–C). Of note, we excluded the in vivo toxicity of Lac against LLC tumor

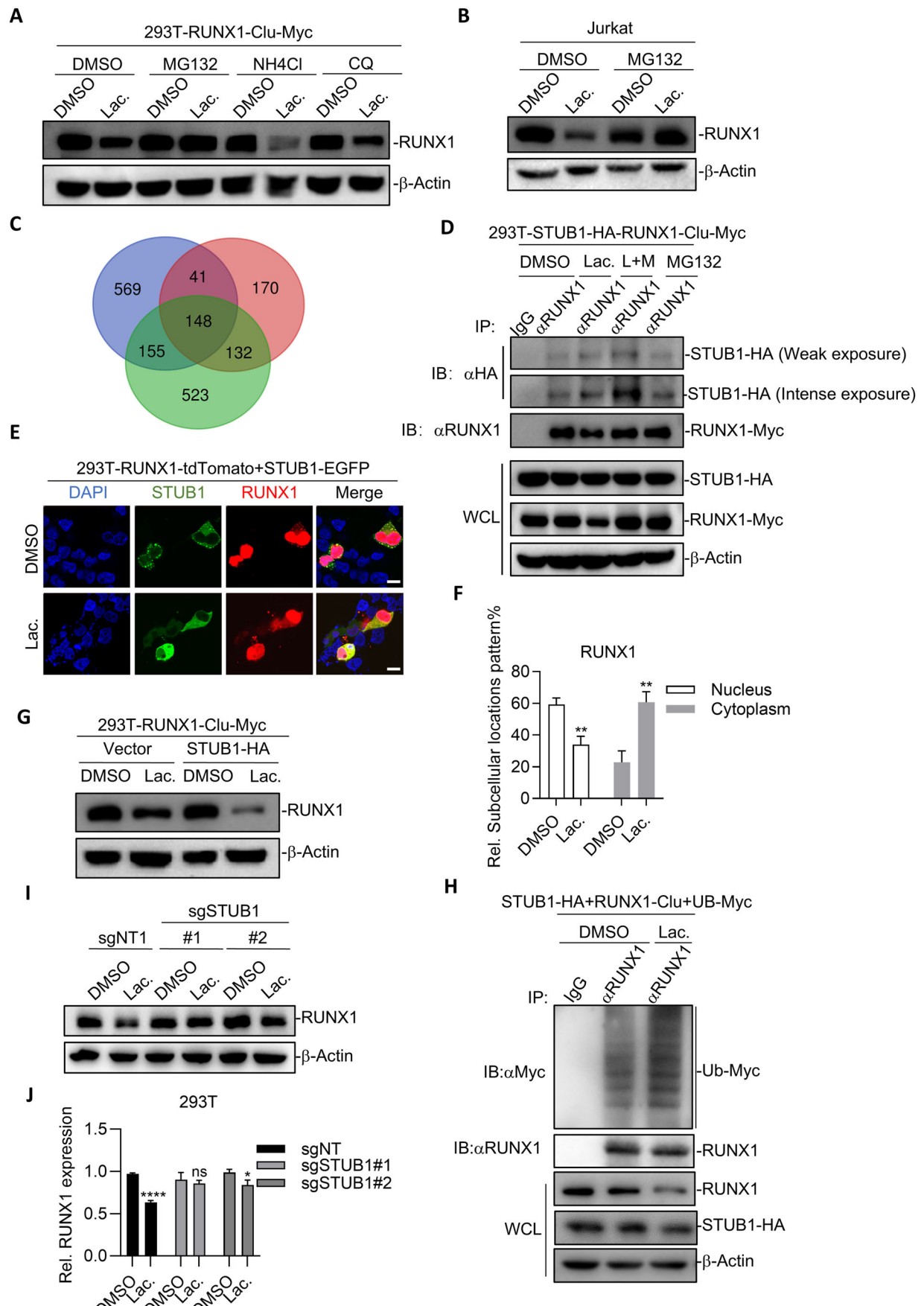

**Figure 4. STUB1 is required for Lac-induced RUNX1 degradation through the proteasome pathway.**

(A, B) Proteasome inhibitor suppresses Lac-induced RUNX1 degradation. 293T-RUNX1-Clu-Myc (A) or Jurkat (B) were pretreated with MG132 (5 μM), NH$_4$Cl (200 ng/mL), or CQ (50 nM) for 1 h, followed by treatment with DMSO or Lac for 6 h. IB was performed with indicated antibodies. (C) Venn diagram of proteins identified in IP-MS. (D) Lac promotes the association between RUNX1 and STUB1. (E) Lac enhances STUB1-mediated cytoplasm redistribution of RUNX1. Scale bar: 10 μm. (F) Statistics analysis of relative subcellular locations pattern of RUNX1 of (E). Data are representative of three independent experiments and were analyzed by Student's *t* test. Error bars denote mean ± SD. *P* value: Nucleus, **$P$ = 0.0027; Cytoplasm, **$P$ = 0.0026. (G) Lac enhances STUB1-mediated degradation of RUNX1. (H) Lac promotes STUB1-induced ubiquitination of RUNX1. (I) STUB1 is required for RUNX1 degradation in the presence of Lac. (J) Statistics analysis of relative RUNX1 expression of (I). Data are representative of three independent experiments and were analyzed by Student's *t* test. Error bars denote mean ± SD. *P* value: sgNT, ****$P$ < 0.0001; sgSTUB1#1, $P$ = 0.4332; sgSTUB1#2, *$P$ = 0.0199. Source data are available online for this figure.

as Lac exhibited no noticeable inhibition of the growth rate of LLC cells at 2 μM in vitro (Appendix Fig. S4A) and did not inhibit the growth of LLC tumor in BALB/c nude mice (Appendix Fig. S4B–D). These data strongly suggested that Lac inhibited LLC tumor growth in B6 mice by enhancing hosts' antitumoral immunity. Importantly, we found that Lac treatment reduced the percentage of Tregs among the tumor-infiltrating CD45$^+$CD4$^+$ population (Fig. 6D,E). Moreover, tumor-infiltrating Tregs were functionally impaired as reflected by reduced CTLA4 expression (Fig. 6F,G).

As autochthonous lung cancers developed in transgenic mouse models better mimic the clinical course of tumorigenesis and development and faithfully recapitulated the complex interaction between tumor and patients' immune system, we generated a cohort of TetO-EGFR (Del19)/CC10rtTA bitransgenic mice (designated EGFR-DEL). These mice developed lung cancer when fed with Doxycycline-containing diet for around 2 months (Ma et al, 2021). To assay the impact of Lac on Tregs in tumor microenvironment, we lethally irradiated these mice, followed by reconstruction of the immune system with bone marrow of FOXP3-DTR-GFP mice (Appendix Fig. S4E,F). We then induced lung cancers in these mice with Dox-diet. Mice were CT-scanned for tumor burden before being randomized for treatment with Lac or diphtheria toxin (DT). As expected, DT treatment had no impact on tumor growth rate in WT EGFR-DEL mice (Fig. 6H,I). Interestingly, we found Lac significantly reduce the tumor burden in WT EGFR-DEL mice (Fig. 6H,I). We also found that both Lac and DT were effective in shrinking lung cancers in bone marrow reconstructed in EGFR-DEL mice (Fig. 6H,I). Pathological examination revealed frequent foci of thickened alveolar wall, intra-tumoral spaces and fibrosis, less Ki67-positive cells in the residual tumors, indicative of healing of tumor areas (Fig. 6J,K). Consistently, flow cytometry analysis revealed that DT deleted Tregs in tumors of bone marrow reconstructed mice, but not of WT non-reconstructed EGFR-DEL mice. Interestingly, we also found that Lac dramatically reduced quantity of tumor-infiltrating Tregs both in the bone marrow reconstructed and non-reconstructed WT EGFR-DEL mice (Fig. 6L,M).

Taken together, our data showed that Lac treatment rewired tumor microenvironment in a Tregs-dependent manner.

### Lac synergizes with PD-1 inhibitors to treat mutant KRAS-driven lung cancer

Mutant KRAS-driven lung adenocarcinoma presents a serious challenge in oncological clinic. We asked whether KRAS mutation-positive lung cancers respond to Lac treatment. To this end, we induced autochthonous lung cancers in our TetO-KRAS$^{G12D}$/

CC10rtTA mice (referred to KRAS$^{G12D}$ mice) following our earlier report (Zhou et al, 2021). Strikingly, Lac treatment dramatically shrank KRAS$^{G12D}$ lung cancer in these mice (Appendix Fig. S5A–D). Consistent with our observation in allograft tumor models, we found that Lac treatment significantly inhibited infiltration of Tregs into tumor loci (Appendix Fig. S5E,F).

KRAS mutation-positive patients respond poorly to treatment with PD-1 inhibitors (Koyama et al, 2016). We therefore set out to test whether Lac synergized with PD-1 inhibitors to treat KRAS$^{G12D}$ lung cancer. Consistent with earlier reports, PD-1 antibody showed negligible therapeutic effect on autochthonous KRAS$^{G12D}$ lung cancers. Strikingly, Lac dramatically enhanced PD-1 antibody to shrink KRAS$^{G12D}$ tumors (Fig. 7A,B). Consistently, we detected a high degree of intra-tumoral space and thickened alveolar walls in tumor loci of mice treated with the combination of PD-1 antibody and Lac (combo treatment), arguing for dramatic remodeling of the area originally occupied by the tumor (Fig. 7A,B). In addition, nuclear morphological heterogeneity and nuclear-cytoplasmic ratio were significantly reduced in combo-treated mice (Fig. 7C,D). Moreover, combo treatment significantly inhibited the proliferation of tumor cells as revealed by Ki67 staining (Fig. 7E,F). We also found that Lac and PD-1 antibody combo treatment reduced Tregs infiltration into tumors (Figs. 7G and EV5A–C). More importantly, residual Tregs in TME were functionally inhibited as revealed by downregulated protein level of CD25, CTLA4, IL10, and PER (Figs. 7G,H and EV5A). In contrast, while Lac or PD-1 antibody treatment did not significantly alter infiltration of CD4$^+$ and CD8$^+$ T cells into tumors (Fig. EV5D,E), these treatments activated tumor-infiltrating T cells as revealed by enhanced expression of IFNγ and perforin (Figs. 7I,J and EV5F,G). These effects were even more dramatic in the combo treatment group (Figs. 7I,J and EV5F,G). Of note, we found no obvious abnormality in the heart, liver, and kidneys of all above-treated mice (Fig. EV5H), showing the safety of Lac in vivo.

## Discussion

Tumor-infiltrating Tregs potently suppress antitumoral response, and thus have long been the target of intensive research for drug development. In our current work, we screened FDA-approved drugs for their ability to inhibit FOXP3/RUNX1 complex formation and identified cardiac glycoside, Lanatoside C, capable of abolishing the maintenance of this complex and inhibiting transcriptional activity of FOXP3. In vitro data showed cardiac glycosides inhibited the proliferation and activity of Tregs. These compounds exhibited potent tumor-shrinking activity in vivo through rewiring tumor immune microenvironment.

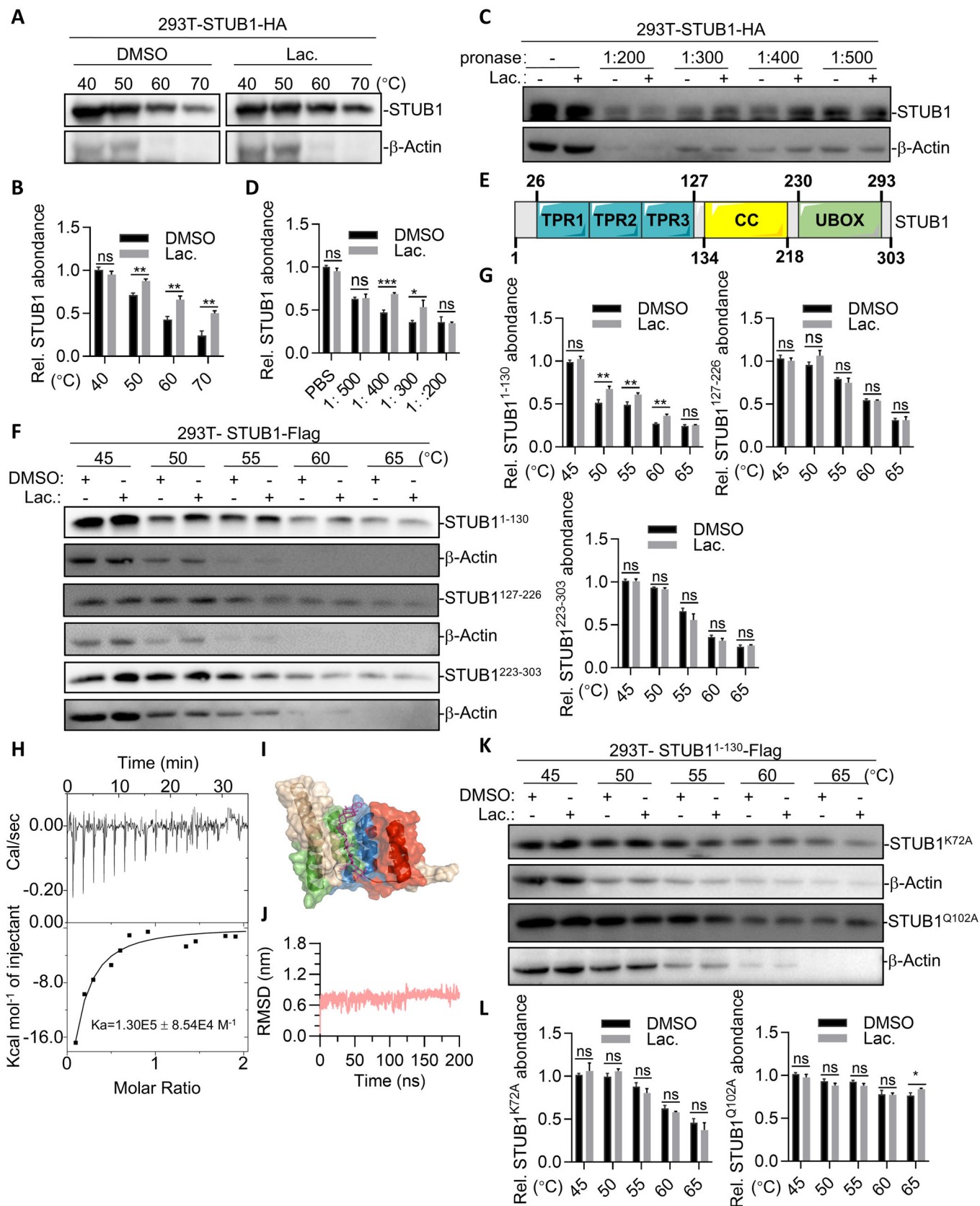

**Figure 5. Lac directly binds STUB1 at its TPR domain to enhance its affinity for RUNX1.**

Bimolecular luciferase complementary assay (Bais et al, 2023) is a convenient method for checking the interaction between two proteins of interest by monitoring luciferase activity, in which the intensity of luciferase directly marks the strength of interaction. This confers a great advantage in high-throughput screening. In our current work, we showed that Lanatoside C dose-dependently inhibited luciferase activity in 293T cells co-expressing Nluciferase-FOXP3 and RUNX1-Cluciferase. Indeed, we have used a similar scheme to assay the ability of chemicals to inhibit the interaction between PD-1 and SHP2 (Fan et al, 2020). In our experience, this method is readily applicable to measure the strength of interaction between proteins of transcription factors as shown in current work here, membrane proteins (Fan et al, 2020) and cytoplasmic proteins (Chen et al, 2020).

Various antibodies are being used or evaluated to deplete Tregs, including those against CD25 (Shimizu et al, 1999), CTLA4 (Selby et al, 2013; Simpson et al, 2013), OX-40 (Bulliard et al, 2014), GITR (Bulliard et al, 2013), and CCR4 (Sugiyama et al, 2013). Given the fact that Tregs share many surface markers with other cell populations, antibody drugs are inevitably limited by their off-target toxicity. By contrast, FOXP3 is the lineage-defining transcription factor for Tregs and thus its specific inhibitors are expected to exert highly effective Tregs inhibitory effect while minimizing off-target toxicities. Recently, antisense oligonucleotide (Revenko et al, 2022), peptide inhibitors (Casares et al, 2010; Hawley et al, 2022; Lozano et al, 2015) and small molecules (Kashima et al, 2016) have been reported to target FOXP3 itself or block the interaction with its partners. However, these reagents need to go through long process before being translated into clinic even if proved effectively. Repurposing an FDA-approved drug, instead, represents a shortcut for developing FOXP3 inhibitor for immediate clinical translation. Our current work exemplifies the successful development of small molecular inhibitors to interfere the transcriptional activity of FOXP3. By repurposing FDA-approved drugs, cardiac glycosides identified here highlight the opportunity for immediate clinical translation in the oncologic clinic.

Although our in vitro data have shown that Lac did not reprogram Tregs into Th1- or Th2-like effector T cells, we can not exclude the possibility of this transforming in vivo. Report shows that attenuated FOXP3 function tends to reprogram Tregs into helper T cells (Wan and Flavell, 2007). In the setting of tumor treatment in vivo, Lac could drastically inhibit the transcriptional

activity of FOXP3. We noticed that Lac treatment reduced the number of Tregs in tumors. Therefore, it's likely that some of Tregs in tumors were reprogrammed into helper T cells by Lac in situ. This would be more advantageous than antibody-mediated depletion of Tregs: helper T cells reprogrammed from Tregs in turn enhance function of CTLs. It's worthy of mentioning a related phenomenon: CD137 agonist therapy reprogramed Tregs into cytotoxic CD4$^+$ T cells with antitumor activity (Akhmetzyanova et al, 2016). It would be even more interesting to check if cardiac glycosides reprogrammed Tregs into cytotoxic CD4$^+$ T cells in vivo.

Tregs accumulate in tumors through three ways: circulating tTregs are recruited into tumor nodules; the conventional CD4$^+$ Foxp3$^-$ (Tconv cells) are converted of into pTregs cells under the influence of tumor-derived factors including TGF-β; and tissue-resident Tregs proliferate on site (Stockis et al, 2019). It will be interesting to study whether these Tregs subsets respond to Lac treatment differently in vivo.

The immunosuppressive tumor microenvironment (TME) represents the major hurdle for successful oncological immunotherapies. Besides Tregs, various cellular components in TME negatively impact on antitumoral response, including myeloid-derived suppressor cells, vascular endothelial cells lymphatic endothelial cells, tumor-associated (TA-) macrophages, and TA-fibroblasts etc. (Balkwill et al, 2012). Frustratingly, some cellular components of both natural and acquired immune arms are coaxed to be immune-suppressive in TME. For example, CD8$^+$ T cell, the major arm responsible for cellular antitumor effect, tends to overexpress CD73 and contribute to immunosuppressive TME through the accumulation of local adenosine (Shevchenko et al, 2020). Another case in point is the dendritic cell (DC), the most important antigen-presenting cell (APC) to educate T cells for fighting cancer. The hypoxic and inflammatory TME transforms DCs to be regulatory DCs (DC-regs) (Shurin et al, 2013). It would be interesting to validate whether Lac likewise inhibited the transcriptional activity of FOXP3 in DC-regs and CD8$^+$ Tregs in vivo.

We have shown here that Lanatoside C warrants clinical translation for treating cancer patients. However, the dose should be carefully titrated in future clinic trials. In our current work, we dosed mice at 6 mg/kg every 2 days. This maybe intolerable to patients. Whether a lower dose of Lanatoside C efficiently rewires the immune microenvironment of cancer patients remains to be determined. With the dazzling

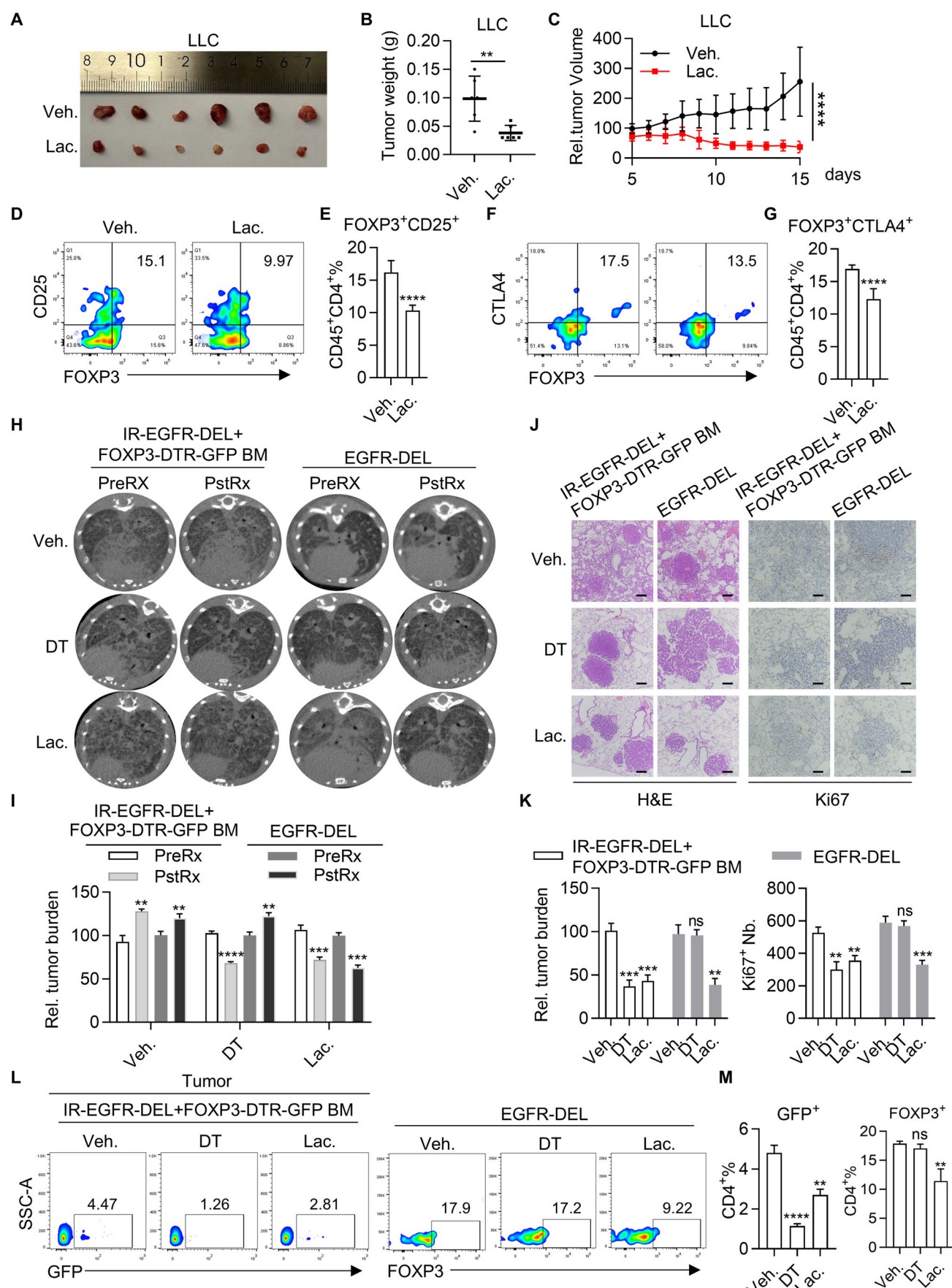

**Figure 6. Lac rewires tumor microenvironment by inhibiting the function of Tregs.**

(A–C) Lac shrinks LLC allograft tumor. The allografts ($n = 6$ mice for each group) were dissected to photograph (A), weigh (B) by the end of experiments. Tumor growth (C) was recorded every day. Values represent the mean ± SD by Student's $t$ test. $P$ value: Rel.tumor weight, $^{**}P = 0.0056$; Rel.tumor volume, $^{****}P < 0.0001$. (D–G) Lac treatment suppresses the infiltration of FOXP3$^+$CD25$^+$ (D, E), FOXP3$^+$CTLA4$^+$ (F, G) Tregs. Data are representative of three independent experiments and were analyzed by Student's $t$ test. Error bars denote mean ± SD. $P$ value: FOXP3$^+$CD25$^+$, $^{****}P < 0.0001$; FOXP3$^+$CTLA4$^+$, $^{****}P < 0.0001$. (H) Lac treatment shrinks autochthonous lung cancer in EGFR-DEL mice. EGFR-DEL mice ($n = 3$ mice for each group) were irradiated with X-ray (left panel) or control (right panel). (I) Statistics of alteration of tumor burden of (H). Data are representative of three independent experiments. $^{**}P < 0.01$, $^{***}P < 0.001$, and $^{****}P < 0.0001$ were analyzed by Student's $t$ test. The exact $P$ values are provided in Dataset EV3. (J) H&E and Ki67 staining of lung sections of mice in (H). Representative images of Hematoxylin and eosin (H&E) (left panel) staining or Ki67 (right panel) expression of the lung tissue obtained from (H). Scale bar: 200 μm. (K) Statistics of relative tumor burden and Ki67-positive cells of (J). Data are representative of three independent experiments. $^{**}P < 0.01$ and $^{***}P < 0.001$ were analyzed by Student's $t$ test. The exact $P$ values are provided in Dataset EV3. (L) Impact of DT or Lac treatment on tumor-infiltrating Tregs population. Representative images of flow cytometry on FOXP3$^+$ or GFP$^+$ population in tumors of mice in (H). (M) Statistics of alteration of GFP$^+$ or FOXP3$^+$ population of (L). Data are representative of three independent experiments and were analyzed by Student's $t$ test. Error bars denote mean ± SD. $P$ value: GFP$^+$ (Veh. vs DT), $^{****}P < 0.0001$; GFP$^+$ (Veh. vs Lac.), $^{**}P = 0.0016$; FOXP3$^+$ (Veh. vs DT), $P = 0.1511$; FOXP3$^+$ (Veh. vs Lac.), $^{**}P = 0.0061$. Source data are available online for this figure.

advances of technologies for targeted delivery of drugs, we could imagine that materials containing Lanatoside C could be implanted within the tumor, or be delivered to tumor loci through complexing with some nanomaterials.

PD-1 antibodies have changed paradigm of clinical treatment of cancer patients. However, only a minority of patients benefit from these drugs. Considering the devastating impact caused by unmet clinical need for effective tumor therapeutics, our finding that cardiac glycosides synergized with PD-1 antibody to shrink tumors sheds new light on possible effective therapy in clinic.

# Methods

### Reagents and tools table

| Reagent/resource | Reference or source | Identifier or catalog number |
|---|---|---|
| CFSE | eBioscience | 65-0850-84 |
| Ovalbumin$_{257-264}$ | Sigma-Aldrich | S7951 |
| Propidium iodide (PI) | Sigma-Aldrich | P4170 |
| **Experimental models** | | |
| OT-I mice | Jackson Lab | 003831 |
| FOXP3-IRES-DTR-GFP mice | Shanghai Model Organisms Center | NM-KI-190046 |
| C57BL/6J | Guangdong Medical Lab Animal Center | |
| BALB/C nude mice | Guangdong Medical Lab Animal Center | |
| TetO-KRASG12D/CC10rtTA mice | Zhou et al, 2021 | |
| TetO-EGFR (Del19)/CC10rtTA mice | Ma et al, 2021 | |
| **Recombinant DNA** | | |
| **Antibodies** | | |
| | Dataset EV2-antibody | Dataset EV2-antibody |
| **Oligonucleotides and other sequence-based reagents** | | |
| QPCR primers | Table EV2-QPCR Primers | Table EV2-QPCR Primers |
| **Chemicals, enzymes, and other reagents** | | |
| Lanatoside C (Lac) | TargetMol | T1670 |
| MG132 | Selleck | S2619 |

| Reagent/resource | Reference or source | Identifier or catalog number |
|---|---|---|
| **Software** | | |
| FlowJo™ v10 | https://www.flowjo.com/ | |
| ImageJ | https://imagej.net/ij/ | |
| Graphpad v10.3.0 | https://www.graphpad.com | |
| **Other** | | |
| Computed tomography | SNC-100, PINGSENG Healthcare | |
| Confocal microscope | Olympus | |
| ITC instrument | ITC-200 | |
| Mass spectrometer | ABI 5600 Triple TOF | |
| Flow cytometry | BD FACSCelesta, BD FACSAria II, BD LSRFortessa | |
| Bright-GloTM Luciferase Assay System | Promega | E2650 |
| His-tag Protein Purification Kit | Beyotime | P2226 |
| CD4$^+$CD25$^+$ regulatory T cell isolation kit | Miltenyi | 130-091-041 |
| MojoSort mouse CD4$^+$ T cell isolation kit | Biolegend | 480005 |
| Annexin V-FITC/PI Apoptosis Detection Kit | Vazyme | A211-02 |
| Mouse IL10 Valukine ELISA Kit | R&D Systems | VAL605 |
| Cytofix/Cytoperm fixation and permeabilization solution | BD Biosciences | 554722 |
| Perm/Wash™ Perm/Wash Buffer | BD Biosciences | 554723 |
| Foxp3/Transcription Factor Staining Buffer Set | eBioscience | 00-5523-00 |

## Plasmid constructs

Plasmid constructs were designed to express FOXP3, RUNX1, STUB1 or STUB1 sgRNA. FOXP3 fusing N-terminal half of firefly luciferase (FOXP3-N-Luc.), RUNX1 fusing C-terminal half of firefly luciferase (RUNX1-C-Luc.), CD27 fusing C-terminal half of firefly luciferase, RUNX1 fusing tdTomato, STUB1 fusing GFP, Flag-tagged FOXP3, Myc-tagged RUNX1, Ub, HA-tagged STUB1, STUB1 truncations and point mutants, His-tagged STUB1[1-130], sgRNA of STUB1 were constructed by standard molecular biology

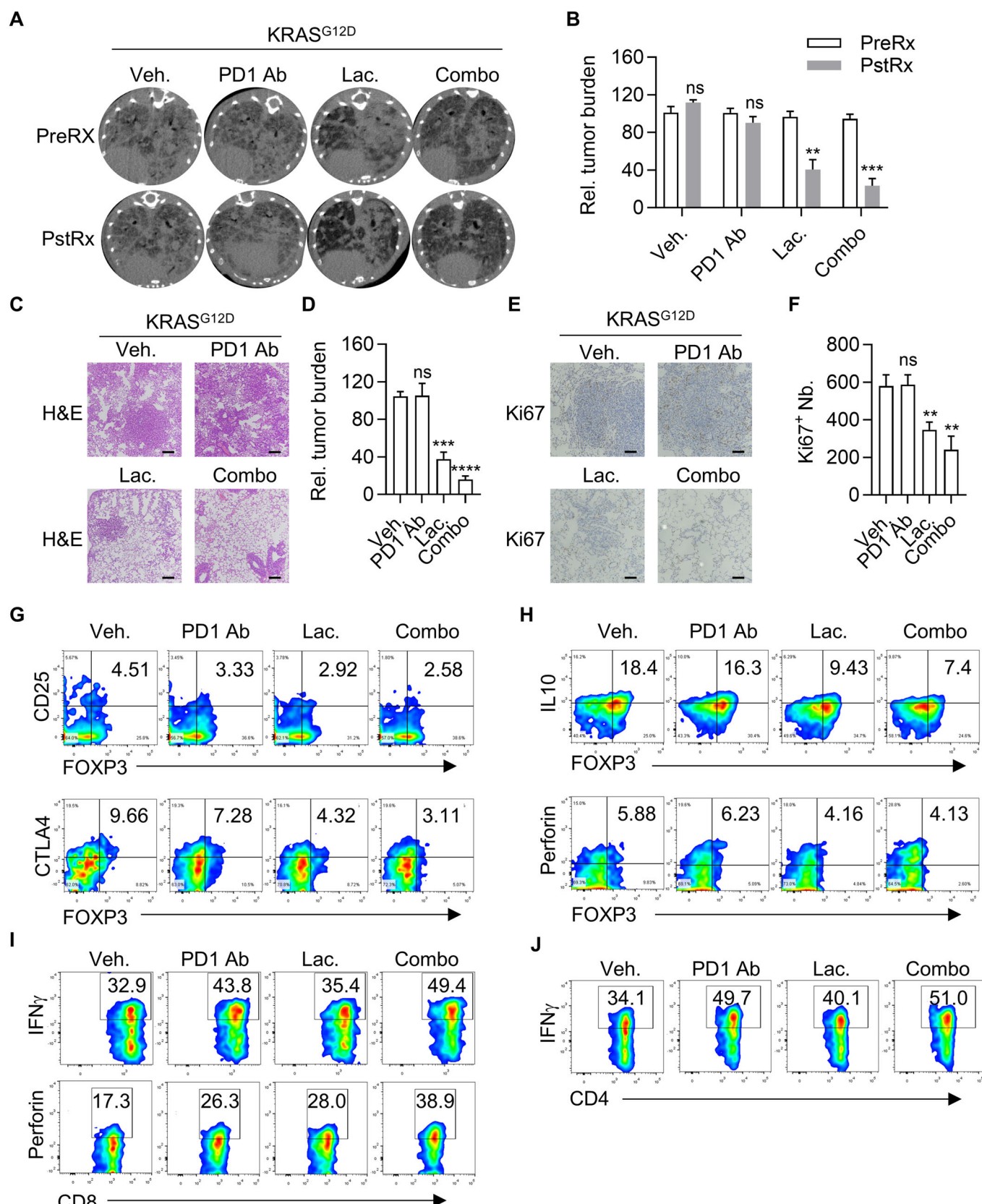

◄ **Figure 7. Lac synergizes with PD-1 inhibitors to treat mutant KRAS-driven lung cancer.**

(A) Lac treatment synergizes with PD-1 antibody to shrink lung cancers. Lung cancer-bearing KRAS[G12D] mice ($n = 3$ mice for each group) were treated with PBS, Lac (6 mg/kg), PD-1 antibody (10 mg/kg) or Lac+PD-1 antibody (Combo) through IP injection. Tumor burden was monitored through CT scanning before and 2-week after treatment. (B) Statistics of tumor burden changes of (A). Data are representative of three independent experiments and were analyzed by Student's $t$ test. Error bars denote mean ± SD. $P$ value: Veh, $P = 0.0630$; PD-1 Ab, $P = 0.0935$; Lac, **$P = 0.0012$; Combo, ***$P = 0.0002$. (C) Representative images of H&E staining of the lung tissues from mice in (A). Scale bar: 200 μm. (D) Statistics of relative tumor burden of (C). Data are representative of three independent experiments and were analyzed by Student's $t$ test. Error bars denote mean ± SD. $P$ value: Veh vs PD-1 Ab, $P = 0.9065$; Veh vs Lac, ***$P = 0.0002$; Veh vs Combo, ****$P < 0.0001$. (E) Representative images of Ki67-positive cells of the lung tissues from different treatment groups. Scale bar: 200 μm. (F) Statistics analysis Ki67-positive cells of (E). Data are representative of three independent experiments and were analyzed by Student's $t$ test. Error bars denote mean ± SD. $P$ value: Veh vs PD-1 Ab, $P = 0.8925$; Veh vs Lac, **$P = 0.0052$; Veh vs Combo, **$P = 0.0033$. (G, H) Impact of drug treatment on tumor-infiltrating Tregs. FOXP3+CD25+ (G, upper panel), FOXP3+CTLA4+ (G, lower panel), FOXP3+IL10+ (H, upper panel), FOXP3+PER+ (H, lower panel) populations in KRAS[G12D] tumors were analyzed by flow cytometry. (I, J) Impact of drug treatment on tumor-infiltrating conventional CD8+ or CD4+ T cells. CD8+IFNγ+ (I, upper panel), CD8+PER+ (I, lower panel), CD4+IFNγ+ (J) populations in KRAS[G12D] tumors were analyzed by flow cytometry. Source data are available online for this figure.

techniques. We used the following plasmid constructs: pCDH-CMV-Nluci-FOXP3-3xFlag-puromycin, pCDH-CMV-RUNX1-Myc-Cluci-zeocin, PCDH-CMV-RUNX1-Myc-tdTomato-zeocin, pCDH-EF1a-STUB1-HA-puromycin, pCDH-EF1a-STUB1-HA-GFP-puromycin, pCDH-EF1a-STUB1[1-130]-3xFlag6xHis-puromycin, pCDH-EF1a-STUB1[127-226]-3xFlag6xHis- puromycin, pCDH-EF1a-STUB1[223-303]-3xFlag6xHis-puromycin, pCDH-EF1a-STUB1[1-130]-K72A-3xFlag6xHis-puromycin, pCDH-EF1a-STUB1[1-130]-Q102A-3xFlag6xHis-puromycin, Lenti-CRISPR-sgSTUB1#1-v2-puromycin, Lenti-CRISPR-sgSTUB1#2-v2-puromycin, pET-T7-STUB1[1-130]-His-laco.

## Flow cytometry and antibodies

Lymph nodes, spleens, or tumors were homogenized and filtrated with nylon mesh. Red blood cells were lysed by red blood cell lysis buffer (RBC). Cells were filtered with strainers to prepare single-cell suspension before staining and flow cytometry. For surface marker staining, the single cells were stained with fluorochrome-conjugated antibodies in PBS containing 1% BSA on ice for 15–20 min according to the manufacturer's instructions. For intracellular staining, the cells were fixed and permeabilized with BD Cytofix/Cytoperm fixation and permeabilization solution or Foxp3/Transcription Factor Staining Buffer Set (eBioscience) at 4 °C according to the manufacturer's instructions. All the FACS antibody details are presented in Dataset EV2.

## Cell lines and culture conditions

Jurkat, LLC, T lymphoma cell line EL4, EG7, HEK293T (293T), and T cells were cultured in incubators at 37 °C and 5% $CO_2$. Jurkat, EL4, and EG7 were cultured in RPMI-1640 containing 10% FBS and 1% PS; LLC and HEK293T cells were cultured in DMEM containing 10% FBS and 1% PS. Splenocytes from OT-I TCR transgenic mice were cultured in RPMI-1640 containing 10 nM of OVA[257-264] peptide, 2-ME (55 μM), and recombinant IL-2 (10 ng/mL). Cells from the spleen and lymph nodes were cultured with precoated αCD3/αCD28 in medium containing IL-2 (10 ng/mL) and 2-ME (55 μM).

## Generation of stable cell line

HEK293T cells were co-transfected with packing plasmids (psPAX2 and pMD2.G, Addgene, #12260 and #12259) and interesting plasmids by PEI. Forty-eight hours after transfection, recombinant virus-containing media was harvested and filtered with 0.45-μm filter. The virus was then added to HEK293T or Jurkat cells in the presence of polybrene (8 μg/mL). After infection, cells were selected using puromycin (1 μg/mL) or Zeocin (330 μg/mL) for at least 7 days before ensuing experiments.

## Mice

OT-I (#003831, Jax); FOXP3-IRES-DTR-GFP mice (FOXP3-DTR-GFP, NM-KI-190046, Shanghai model organisms center); C57BL/6J, BALB/C nude mice (Guangdong medical lab animal center); TetO-KRAS[G12D]/CC10rtTA (KRAS[G12D]) and TetO-EGFR (Del19)/CC10rtTA mice (EGFR-DEL) transgenic mice were maintained on C57BL/6J background. All mice were kept in the Laboratory Animal Science Institute, Jinan University. All animal work was performed in accordance with the approved protocol. All animal protocols and procedures were approved by the Animal Use and Care Committee of Jinan University (approved number: 101266).

## Thermal shift assay

293T cells ($5 \times 10^6$) were transfected with indicated plasmids for 36 h. Cell transformants were harvested and lysed with NP40 on ice for 30 min. The whole-cell lysate was centrifuged to remove cell debris. The supernatant was incubated with DMSO or Lac (10 μM) for 30 min, and subsequently treated at different temperature gradients for 3 min. The lysates were then centrifuged to remove denatured proteins and the supernatant was analyzed through IB.

## DARTS (drug affinity responsive target stability)

293T cells ($5 \times 10^6$) were transfected with indicated plasmids for 36 h. The cells were harvested and lysed with NP40 on ice for 30 min. Whole-cell lysate was centrifuged to remove cell debris. Supernatants were incubated with DMSO or Lac (30 μM) for 3 h, and subsequently treated with pronase of indicated concentration for 20 min before IB analysis.

## Confocal microscopy

293T cells ($1 \times 10^5$) were transfected with indicated plasmids. Thirty-six hours after transfection, cells were pretreated with MG132 for 1 h, followed by treatment with DMSO or Lac (2 μM) for 4 h. The cells were fixed with 4% paraformaldehyde for 20 min and stained with DAPI before confocal microscopy analysis.

## Co-immunoprecipitation (Co-IP) and immunoblot (IB)

293T-NF + RC cells ($5 \times 10^6$) or 293T cells ($5 \times 10^6$) were co-transfected with indicated plasmids for around 36 h. The transformants were treated with reagents indicated in the manuscript and then lysed. Co-IP and IB were performed on lysates, as described previously (Zhou et al, 2022).

## Protein purification

BL21 was transformed with pET-T7-STUB1$^{1-130}$-His and cultured for 5 h. BL21 transformants were then treated with IPTG (1 mM) for 4 h. The supernatant was harvested and purified by His-tag protein purification kit. The protein was concentrated before further experiments.

## Isothermal titration calorimetry (ITC)

The purified STUB1$^{1-130}$ protein was adjusted to 8 μM and Lac to 80 μM with PBS buffer. Drug was added dropwise into the protein solution, controlled by blank PBS buffer into protein solution. Heat reads of samples were subtracted with that of the control experiment. Kd and Ka were calculated with software (One set of sites).

## Molecular dynamics simulation

Molecular docking was performed with CDOCER program. Crystal structure of TPR domain of STUB1 protein was retrieved from PDB database (PDB ID: 8F15). Molecular dynamics simulation of the docked conformation of Lac/STUB1 was conducted by GROMACS program for 200 ns. Ligplot software was used to analyze the interaction between the Lac and TPR domain of the STUB1. Free energy for binding between Lac and STUB1 was calculated with Discovery studio 2021.

## Real-time PCR

Total RNA was isolated from cells using Trizol reagent (TAKARA, Japan) before real-time PCR quantifying mRNA levels of the indicated genes. Data shown are the relative abundance of the indicated mRNA normalized to that of *GAPDH* or *β-Actin*. Gene-specific real-time PCR primer sequences as shown in Table EV2.

## Luciferase assay

293T-NF + RC ($1 \times 10^5$) cells were seeded in a 96-well plate overnight. After treatment with drugs, Bright-Glo™ Luciferase Assay System was then applied to detect luciferase activity.

## Silver staining and mass spectrometry analysis

293T-RUNX1-Clu-Myc cells ($3 \times 10^7$) were pretreated with MG132 (5 μM) for 1 h, followed by treatment with Lac for 6 h. Subsequently, immunoprecipitants with RUNX1 antibody or IgG were subjected to IP-MS analysis in three biological replicates. Silver staining was performed according to rapid silver staining kit's protocol. The strip was divided into several components (strip width is about 2 mm) followed by in-gel digestion (Zhou et al,

2021). The product of in-gel digestion was applied to the mass spectrometry assays.

## Re-immunoprecipitation and ubiquitination assay

Cells were lysed in lysis buffer containing 1% SDS and immunoprecipitated with αRUNX1. Immunoprecipitants was denatured by heating for 5 min. The concentration of SDS in immunoprecipitants was adjusted with NP40 to 0.1%. The diluted supernatants were re-immunoprecipitated with αRUNX1. The immunoprecipitants were then analyzed by IB with antibody against ubiquitin (Zhong et al, 2009).

## Preparations and differentiation of mouse Tregs cells

Tregs cells were separated from lymph nodes by CD4$^+$CD25$^+$ regulatory T cell isolation kit with magnetic-activated cell sorting (MACS) or Fluorescence-activated cell sorting (FACS). The purified Tregs were cultured with precoated αCD3/αCD28 in medium containing 2-ME (55 μM) and recombinant IL-2 (10 ng/mL) before further experiments.

CD4$^+$ T cells were separated from lymph nodes with MojoSort mouse CD4$^+$ T cell isolation kit according to manufacturer's instructions. The purified CD4$^+$ T cells were cultured with precoated αCD3/αCD28 in RPMI-1640 containing 2-ME (55 μM), recombinant IL-2 (10 ng/mL) and recombinant TGF-β (10 ng/mL) for 6 days to promote Tregs differentiation.

## Impact of Tregs on cytotoxicity of OT-I CTLs

Splenocytes isolated from OT-I mice were stimulated with OVA$_{257-264}$ peptide (10 nM) for 72 h in the presence of IL-2 (10 ng/ml) and 2-ME (55 μM) to generate mature CTLs. Tregs were treated with DMSO or Lac (2 μM) for 6 h in advance. EG7 cells were stained with CFSE. The mature CTLs, pretreated Tregs, and CFSE-labeled EG7 (Tregs:CTLs:EG7 = 10:5:2) were co-cultured in the presence of DMSO or Lac (2 μM) for 6 h. The cells were then stained with PI before flow cytometry analysis. Killing of target cells were determined by analyzing PI/CFSE double-positive population.

## Impact of Tregs on the proliferation of OT-I CTLs

Purified Tregs cells were cultured with precoated αCD3/αCD28 in RPMI-1640 containing 2-ME (55 μM) and recombinant IL-2 (10 ng/ml), followed by treatment with DMSO or Lac for 6 h. These Tregs were co-incubated with CFSE-labeled OT-I lymphocytes (Tregs:OT-I lymphocytes = 5:1) in the presence of DMSO or Lac (2 μM) for 48 h. Proliferation of OT-I cells were determined by dilution of CFSE through flow cytometry analysis on CD8a positive population. Division index (DI) and proliferation index (PI) were calculated by the FlowJo software.

## Impact of Tregs on cytokine production by OT-I CTLs

Splenocytes isolated from OT-I mice were stimulated with OVA$_{257-264}$ peptide (10 nM) for 72 h in the presence of IL-2 (10 ng/ml) and 2-ME (55 μM) to generate mature CTLs. Tregs were treated with DMSO and Lac (2 μM) for 6 h in advance. The mature CTLs, pretreated Tregs and

EG7 (Tregs:CTLs:EG7 = 10:5:2) were co-cultured in the presence of DMSO or Lac (2 μM) for 6 h, and then monensin was added and incubated for another 6 h. The cells were stained with CD8a, IFNγ, Perforin, and GRZB followed by flow cytometry analysis.

## Analyses of apoptosis of Tregs

Purified Tregs were cultured with plate-coated αCD3/αCD28 (1 μg/mL), 2-ME (55 μM) and recombinant IL-2 (10 ng/ml), followed by treatment with DMSO or Lac for 6 h. These Tregs were stained with Annexin V-FITC and PI according to the manufacturer's instructions before flow cytometry analysis.

## Depletion of Tregs in vivo

The EGFR-DEL mice ($n = 3$ mice for each group, female) were lethally irradiated with X-ray (8 Gy for 20 min) or shank. Irradiated mice were transplanted with $5 \times 10^6$ of bone marrow cells from FOXP3-DTR-GFP mice. Three days later, mice were fed with doxycycline diet to induce lung cancer for 2 months. Mice were randomized for treatment with PBS, DT or Lac every 2 days for 2 weeks. Tumor burdens were monitored through CT imaging before and after treatment. The efficiency of depletion of Tregs was evaluated by flow cytometry.

## Xenograft model in vivo

LLC cells ($5 \times 10^5$) were resuspended in mixture of equal volume of Matrigel and PBS. 6-week-old wild-type C57BL/6 (female) or BALB/C (female) nude mice were inoculated with LLC cells mixture subcutaneously on the right flank. When tumors grew to a volume of around 60 mm³, mice were randomized for treatment ($n = 6$) with PBS or Lac (6 mg/kg) via intraperitoneal injection (IP) every 2 days for 2 weeks. Tumor volumes were monitored at indicated time points. At the end of treatment, the mice were euthanized and tumor allografts were dissected, weighed, and photographed.

LLC allograft tumor was used for evaluating the infiltration of Tregs by flow cytometry.

## Transgenic tumor model in vivo

One-month-old KRAS^[G12D] or EGFR-DEL bitransgenic mice ($n = 3$ mice for each group, female) were fed with doxycycline-containing diet for around 8–12 weeks to induce lung cancer. Tumor burdens were confirmed by computed tomography before being randomized for treatments with Lac (6 mg/kg, IP), PD-1 antibody (10 mg/kg, IP) or combination treatments every 2 days for 2 weeks, respectively. Mice were CT-scanned for monitoring tumor volume before sacrifice. Part of lung tissues were fixed for pathological analysis. Parallelly, another part of lungs was used to harvest tumor nodules for flow cytometry analysis.

## Hematoxylin and eosin (H&E) staining, immunohistochemical analysis (IHC), or immunofluorescence

After the euthanasia of mice, tumor allograft, lung, heart, kidney, and liver tissues were dissected from the mice and fixed overnight in 10% formalin solution. The tissues were dehydrated, embedded in paraffin, sectioned and stained with H&E, αKi67 or indicated antibodies.

### The paper explained

#### Problem
The immune-suppressive Tregs in tumor microenvironment represent a formidable hurdle for successful immunotherapy. Unfortunately, Treg-specific inhibitors are not yet available in oncological clinics. FOXP3, the lineage-defining transcription factor for Tregs, plays an important role in the function and maintenance of Tregs. For its transcriptional activity, FOXP3 must form a complex with RUNX1. To explore the potential repurposing of FDA-approved drugs for tumor immunotherapy, we screened FDA-approved drugs for their ability to disrupt the complex of FOXP3/RUNX1.

#### Results
We identified an FDA-approved drug, Lanatoside C (Lac), capable of efficiently promoting STUB1 E3 ligase to bind and ubiquitylate RUNX1 for degradation, thus disrupting the complex of FOXP3/RUNX1 and abandoning the transcription of genes necessary for immunosuppressive function of Tregs. Lac potently inhibits the immunosuppressive function of Tregs in vitro and efficiently regresses tumors in vivo.

#### Impact
Our work pointed to a potential immediate clinical translation of Lanatoside C for tumor treatment, either alone or in combination with PD-1 inhibitors. Our work also shed light on a new strategy for developing FOXP3 inhibitors.

## Statistical analysis

Statistics were performed with GraphPad Prism 10.1.0. Student's $t$ test was used to compare differences between the two experimental groups. Data are presented as the mean ± SD, and error bars denote SD; $n = 3$; *$P < 0.05$; **$P < 0.01$; ***$P < 0.001$; ****$P < 0.0001$. All the $P$ value was presented in Dataset EV3. ImageJ was used to analyze the gray value of IB.

## Data availability

The mass spectrometry proteomics data have been deposited to the ProteomeXchange Consortium via the PRIDE (Perez-Riverol et al, 2022) partner repository with the dataset identifier PXD059236. Any data and materials used in this analysis can be made available on request to the corresponding author except as restricted by material transfer agreements (MTAs).

The source data of this paper are collected in the following database record: biostudies:S-SCDT-10_1038-S44321-025-00200-y.

## Peer review information

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

## Acknowledgements

The authors gratefully acknowledge financial support for this research from the National Key Research and Development Program of China (2022YFA1103900), Key Technologies R&D Program of Guangdong Province (2023B1111030003), the National Natural Science Foundation of China (32370966), the Guangdong Basic and Applied Basic Research Foundation (2024A1515030238), the Guangzhou Basic Research Program (2024A03J0596, 2025A04J5238), the Open Project Funded by the MOE Key Laboratory of Tumor Molecular Biology (202302).

## Author contributions

Qian Zhou: Data curation; Software; Formal analysis; Supervision; Funding acquisition; Validation; Investigation; Visualization; Methodology; Writing— original draft; Project administration; Writing—review and editing. **Tong Yang**: Data curation; Methodology. **Xixi Yu**: Data curation; Software; Formal analysis; Investigation; Visualization; Methodology; Writing—original draft. **Bo Li**: Data curation; Software; Formal analysis; Funding acquisition; Validation; Investigation; Visualization; Methodology; Writing—original draft. **Jin Liu**: Data curation; Software; Formal analysis; Validation; Investigation; Methodology. **Yongxin Mao**: Data curation; Formal analysis; Methodology. **Rongxiang Guo**: Data curation; Software; Formal analysis; Investigation; Methodology. **Zhuo Feng**: Data curation; Software; Investigation. **Li Zhou**: Data curation; Software; Investigation; Methodology. **Guandi Zeng**: Data curation; Investigation; Methodology. **Nan Li**: Data curation; Supervision; Investigation. **Jinxia Liang**: Data curation; Formal analysis; Methodology. **Lu Liu**: Data curation; Formal analysis; Methodology. **Pengju Feng**: Data curation; Investigation; Methodology. **Hong-Bing Shu**: Conceptualization; Resources; Supervision; Funding acquisition; Visualization; Writing—original draft; Project administration; Writing— review and editing. **Liang Chen**: Conceptualization; Resources; Supervision; Funding acquisition; Visualization; Writing—original draft; Project administration; Writing—review and editing.

Source data underlying figure panels in this paper may have individual authorship assigned. Where available, figure panel/source data authorship is listed in the following database record: biostudies:S-SCDT-10_1038-S44321-025-00200-y.

## Disclosure and competing interests statement

The authors declare no competing interests.

# Expanded View Figures

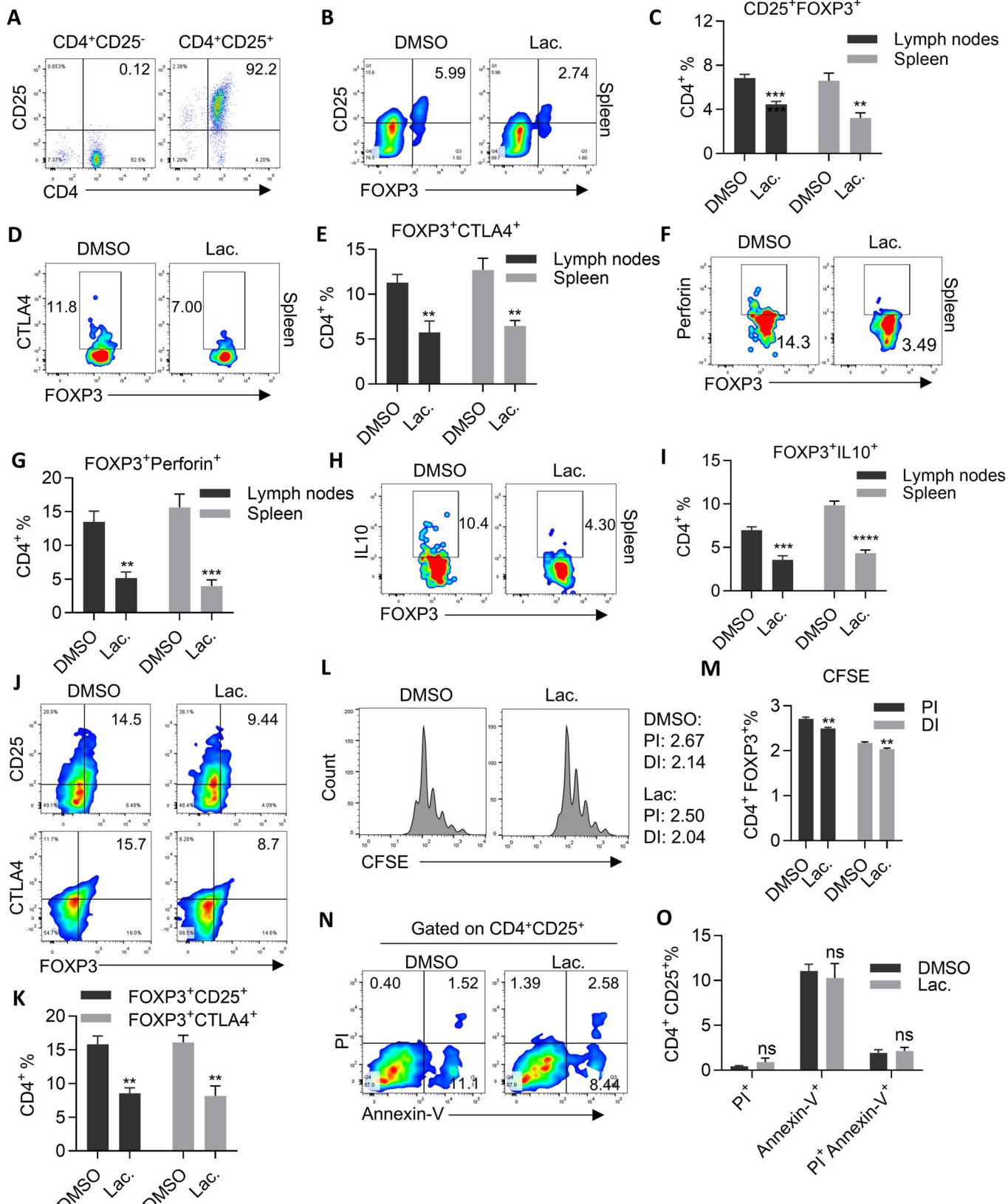

**Figure EV1.  Impact of Lac on Tregs.**

(A) Tregs-purifying efficiency from mouse lymph nodes. Conventional CD4$^+$ T (CD4$^+$CD25$^-$) or Treg (CD4$^+$CD25$^+$) cells were purified from lymph nodes by negative selection with MACS before flow cytometry assay. (B–I) Flow cytometry analysis of the impact of Lac on marker protein expression by Tregs. The gating of FOXP3$^+$CD25$^+$ (B), FOXP3$^+$CTLA4$^+$ (D), FOXP3$^+$Perforin$^+$ (F) and IL10$^+$ (H) populations were determined against their FMO; Statistics analysis of Fig. 1D and EV1B (EV1C), Fig. 1E and EV1D (EV1E), Fig. 1F and EV1F (EV1G), Fig. 1G and EV1H (EV1I). Data are representative of three independent experiments and were analyzed by Student's *t* test. Error bars denote mean ± SD. *P* value: FOXP3$^+$CD25$^+$ (Lymph nodes), ***P* = 0.0006; FOXP3$^+$CD25$^+$ (Spleen), ***P* = 0.0024; FOXP3$^+$CTLA4$^+$ (Lymph nodes), ***P* = 0.0035; FOXP3$^+$CTLA4$^+$ (Spleen), ***P* = 0.0017; FOXP3$^+$Perforin$^+$ (Lymph nodes), ***P* = 0.0014; FOXP3$^+$Perforin$^+$ (Spleen), ***P* = 0.0008; FOXP3$^+$IL10$^+$ (Lymph nodes), ***P* = 0.0006; FOXP3$^+$IL10$^+$ (Spleen), ****P* < 0.0001. (J) Lac suppresses Tregs differentiation in vitro. Representative flow plots were gated on CD4$^+$ population for analysis. The FOXP3$^+$CD25$^+$ (upper panel) or FOXP3$^+$CTLA4$^+$ (lower panel) populations were determined against their FMO, respectively. (K) Statistics analysis of Fig. EV1J. Data are representative of three independent experiments and were analyzed by Student's *t* test. Error bars denote mean ± SD. *P* value: FOXP3$^+$CD25$^+$, ***P* = 0.0010; FOXP3$^+$CTLA4$^+$, ***P* = 0.0016. (L) Lac suppresses the proliferation of Tregs in vitro. Proliferation of Tregs were determined by dilution of CFSE through flow cytometry analysis. (M) Statistics of the percentages of proliferating Tregs of Fig. EV1L. Data are representative of three independent experiments and were analyzed by Student's *t* test. Error bars denote mean ± SD. *P* value: PI, ***P* = 0.0014; DI, ***P* = 0.0041. (N) Impact of Lac on apoptosis of Tregs. Cells were stained with PI and Annexin V-FITC for flow cytometry analysis. (O) Statistics of apoptosis population of Fig. EV1N. Data are representative of three independent experiments and were analyzed by Student's *t* test. Error bars denote mean ± SD. *P* value: PI$^+$, *P* = 0.1653; Annexin V$^+$, *P* = 0.4850; Annexin V$^+$PI$^+$, *P* = 0.4890. Source data are available online for this figure.

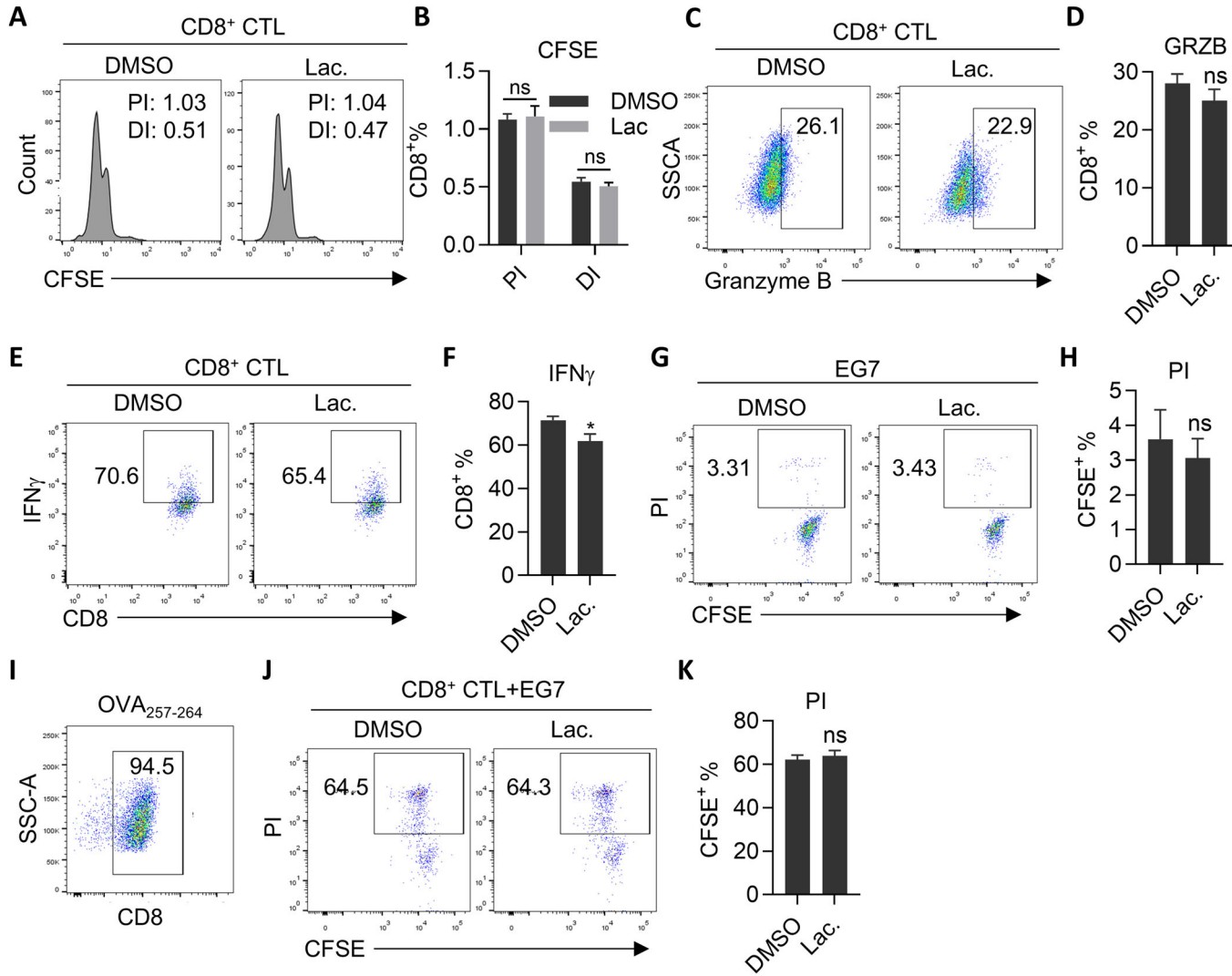

**Figure EV2.    Impact of Lac on the phenotype of CD8+ T cells.**

(A) Lac exhibits limited impact on the proliferation of CD8+ T cells. Proliferation of OT-I cells were determined by dilution of CFSE through flow cytometry analysis on CD8a positive population. (B) Statistics of percentages of proliferating OT-I CTLs of Fig. EV2A. Data are representative of three independent experiments and were analyzed by Student's *t* test. Error bars denote mean ± SD. *P* value: PI, *P* = 0.6702; DI, *P* = 0.2193. (C–F) The mature CTLs and EG7 cells (CTLs:EG7 = 5:2) were co-cultured in the presence of DMSO or Lac (2 μM) for 6 h, followed by incubation with monensin for another 6 h. The cells were then stained with anti-CD8a and GRZB (EV2C) or anti-CD8a and IFNγ (EV2E) for flow cytometry analysis. Statistics of GRZB (EV2D) and IFNγ (EV2F) expression. Data are representative of three independent experiments and were analyzed by Student's *t* test. Error bars denote mean ± SD. *P* value: GRZB, *P* = 0.1202; IFNγ, *P* = 0.0114. (G) Lac exhibits no toxicity to EG7 cells. Toxicity of EG7 cells was determined by analyzing PI/CFSE double-positive population. (H) Statistics of the percentages of CFSE+PI+ EG7 cells of Fig. EV2G. Data are representative of three independent experiments and were analyzed by Student's *t* test. Error bars denote mean ± SD. *P* value: *P* = 0.4103. (I) Purity of OVA$_{257-264}$ peptide-induced splenic OT-I CTLs. (J) Impact of Lac treatment on cytotoxicity of CTLs. (K) Statistics of the percentages of PI/CFSE double-positive EG7 cells of Fig. EV2J. Data are representative of three independent experiments and were analyzed by Student's *t* test. Error bars denote mean ± SD. *P* value: *P* = 0.3614. Source data are available online for this figure.

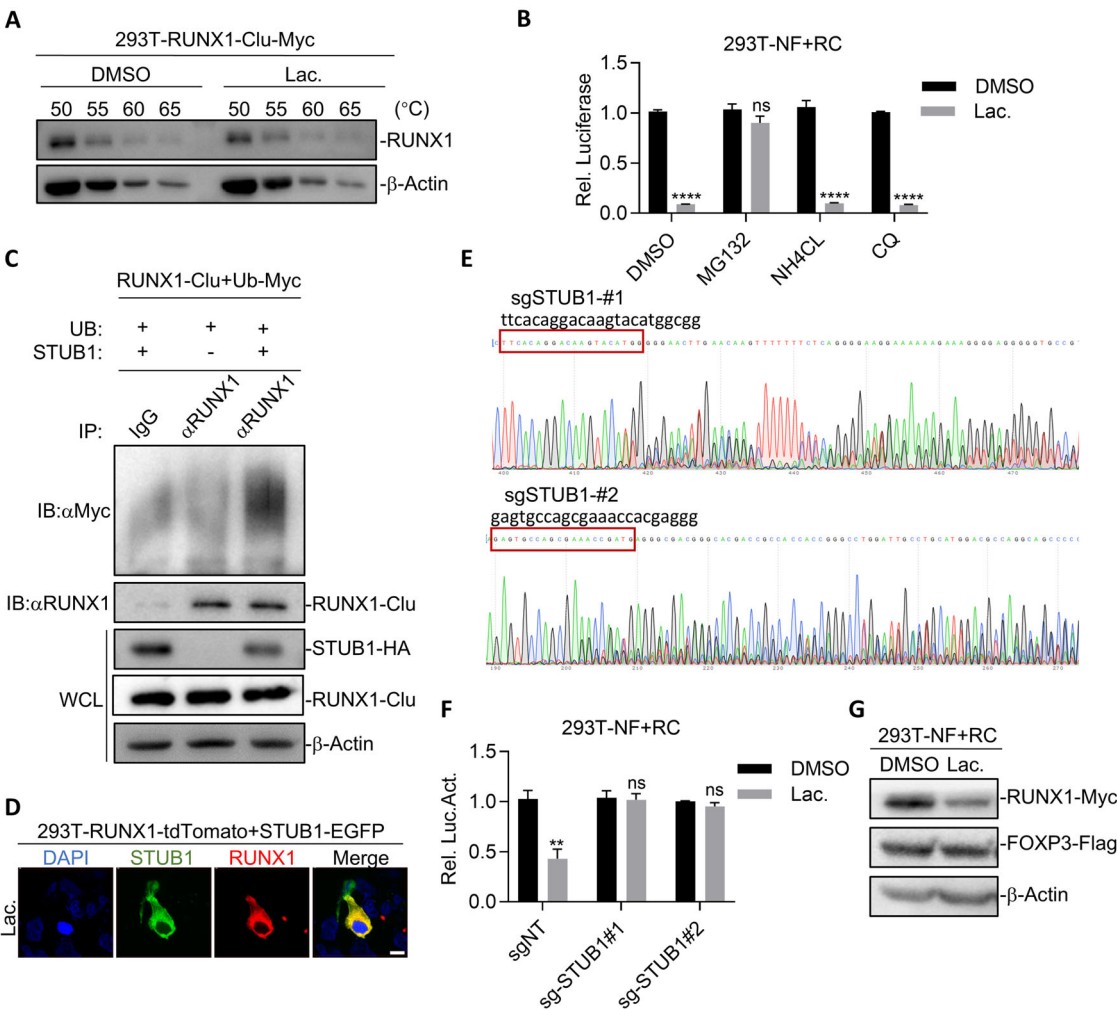

**Figure EV3. STUB1 is required for Lac-induced RUNX1 degradation through proteasome pathway.**

(A) Lac doesn't change the thermal stability of ectopically expressed RUNX1. (B) MG132, but not NH4Cl or CQ, stabilizes luciferase activity in response to Lac treatment in 293T-NF + RC cells. Data are representative of three independent experiments and were analyzed by Student's t test. Error bars denote mean ± SD. P value: DMSO, ****P < 0.0001; MG132, P = 0.0537; NH4CL, ****P < 0.0001; CQ, ****P < 0.0001. (C) STUB1-mediated induction of ubiquitination of RUNX1. (D) Lac promotes STUB1-mediated redistribution of RUNX1. Scale bar: 10 μm. (E) Sequence chromatograms of 293T-sgSTUB1 cells. (F) STUB1 knockout stabilizes the luciferase activity of 293T-NF + RC in response to Lac treatment in bimolecular fluorescence complementation assay. Data are representative of three independent experiments and were analyzed by Student's t test. Error bars denote mean ± SD. P value: sgNT, **P = 0.0013; sgSTUB1#1, P = 0.7501; sgSTUB1#2, P = 0.0811. (G) Lac promotes the degradation of RUNX1 but not FOXP3 in 293T-NF + RC cells. Source data are available online for this figure.

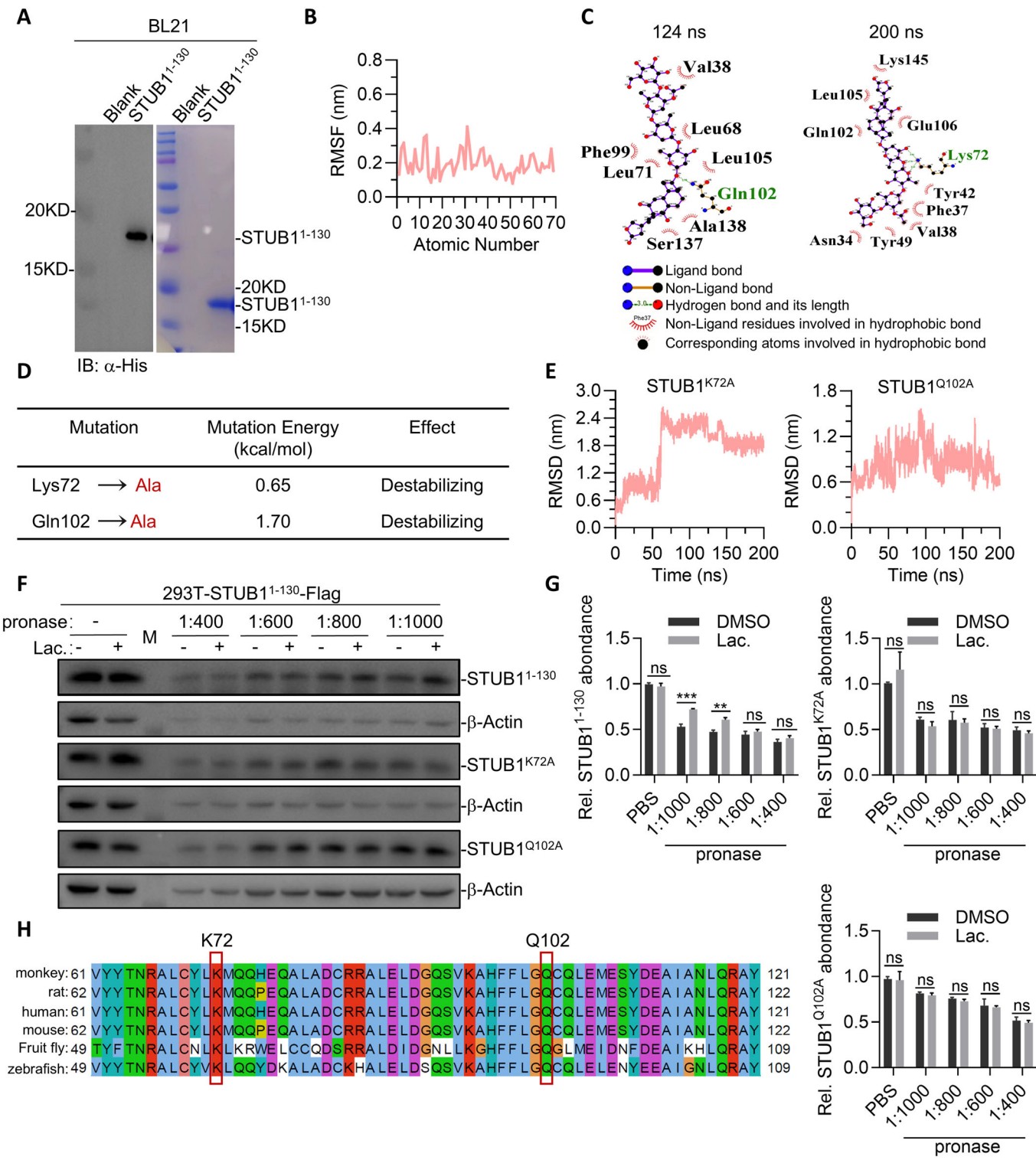

◄ **Figure EV4. Lac directly binds STUB1 at its TPR domain to enhance its affinity to RUNX1.**

(A) IB and coomassie brilliant blue staining of purified STUB1$^{1-130}$. IB (left panel) and SDS-PAGE (right panel) analysis of purified STUB1$^{1-130}$ protein. (B) The root-mean-square deviations (RMSF) plot for docked Lac during MD simulations. (C) Ligplot of 2-D interaction of Lac-STUB1 at 124 ns and 200 ns in the MD simulations (The red arcs for hydrophobic interactions; green lines for hydrogen bonds). (D) Free energies predicted for bindings of Lac-STUB1$^{K72A}$ (Lys72 to Ala) and Lac-STUB1$^{Q102A}$ (Gln102 to Ala) with Lac. (E) Time-dependent RMSD plot of docked Lac in single-point mutants of STUB1 (K72A and Q102A) during molecular dynamics simulation. (F) Lac fails to protect ectopically expressed mutant STUB1 from pronase digestion through DARTS assay. (G) Statistics of relative abundance of mutant STUB1 of (F). Data are representative of three independent experiments and were analyzed by Student's $t$ test. Error bars denote mean ± SD. $P$ value: PBS, $P = 0.3196$, 1: 1000, ***$P = 0.0005$, 1: 800, **$P = 0.0021$, 1: 600, $P = 0.2729$, 1: 400, $P = 0.1308$ for STUB1$^{1-130}$; PBS, $P = 0.2618$, 1: 1000, $P = 0.0892$, 1: 800, $P = 0.6736$, 1: 600, $P = 0.5848$, 1: 400, $P = 0.2594$ for STUB1$^{127-226}$; PBS, $P = 0.8264$, 1: 1000, $P = 0.3046$, 1: 800, $P = 0.1404$, 1: 600, $P = 0.7153$, 1: 400, $P = 0.4499$ for STUB1$^{223-303}$. (H) Sequence homology of STUB1 TPR domain. Alignment of STUB1 sequences from human (H. sapiens, 005852.2), zebrafish (D. rerio, 021325787.1), mouse (M. musculus, 062693.1), rat (R. norvegicus, 001020796.2), monkey (M. mulatta, 001244487.1), Fruit fly (D. melanogaster, 477441.1). Source data are available online for this figure.

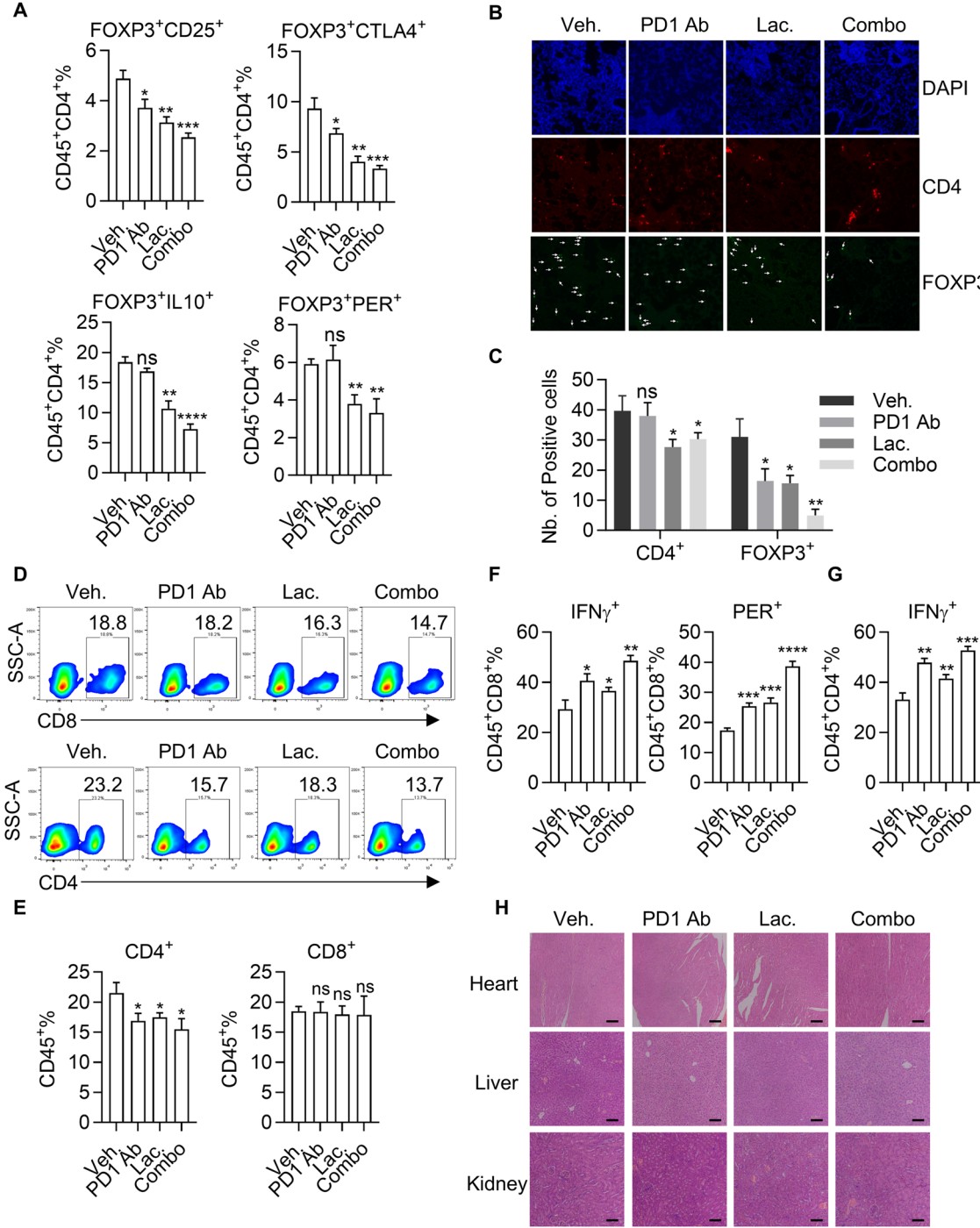

**◄**

**Figure EV5.  Lac synergizes with PD-1 inhibitor to treat mutant KRAS-driven lung cancer.**

(A) Statistics of the proportion of FOXP3⁺CD25⁺, FOXP3⁺CTLA4⁺, FOXP3⁺IL10⁺ and FOXP3⁺PER⁺ Tregs in tumor of Fig. 7G, H. Data are representative of three independent experiments and were analyzed by Student's *t* test. Error bars denote mean ± SD. *P* value: Veh. vs PD-1 Ab, \**P* = 0.0127, Veh. vs Lac, \*\**P* = 0.0015, Veh. vs Combo, \*\*\**P* = 0.0004 for FOXP3⁺CD25⁺; Veh. vs PD-1 Ab, \**P* = 0.0218, Veh. vs Lac, \*\**P* = 0.0016, Veh. vs Combo, \*\*\**P* = 0.0007 for FOXP3⁺CTLA4⁺; Veh. vs PD-1 Ab, *P* = 0.0624, Veh. vs Lac, \*\**P* = 0.0010, Veh. vs Combo, \*\*\*\**P* < 0.0001 for FOXP3⁺IL10⁺; Veh. vs PD-1 Ab, *P* = 0.6317, Veh. vs Lac, \*\**P* = 0.0027, Veh. vs Combo, \*\**P* = 0.0049 for FOXP3⁺PER⁺. (B) Immunofluorescence analysis of Tregs infiltration. Scale bar: 100 μm. (C) Statistics of the Tregs infiltration (CD4⁺; FOXP3⁺) in tumor of Fig. EV5B. Data are representative of three independent experiments and analyzed by Student's *t* test. Error bars denote mean ± SD. *P* value: Veh. vs PD-1 Ab, *P* = 0.6870, Veh. vs Lac, \**P* = 0.0210, Veh. vs Combo, \**P* = 0.0412 for CD4⁺; Veh. vs PD-1 Ab, \**P* = 0.0246, Veh. vs Lac, \**P* = 0.0151, Veh. vs Combo, \*\**P* = 0.0021 for FOXP3⁺. (D) Lac or PD-1 antibody treatment did not significantly alter the infiltration of CD4⁺ and CD8⁺ T cells into tumors. Representative flow plots were gated on CD45⁺ population for analysis. (E) Statistics of the proportion of CD8⁺ (right panel) and CD4⁺ (left panel) in tumor of Fig. EV5D. Data are representative of three independent experiments and were analyzed by Student's *t* test. Error bars denote mean ± SD. *P* value: Veh. vs PD-1 Ab, \**P* = 0.0191, Veh. vs Lac, \**P* = 0.0195, Veh. vs Combo, \**P* = 0.0129 for CD4⁺; Veh. vs PD-1 Ab, *P* = 0.9277, Veh. vs Lac, *P* = 0.5793, Veh. vs Combo, *P* = 0.7526 for CD8⁺. (F) Statistics of the proportion of CD8⁺IFNγ⁺ (left panel) and CD8⁺PER⁺ (right panel) in tumor of Fig. 7I. Data are representative of three independent experiments and were analyzed by Student's *t* test. Error bars denote mean ± SD. *P* value: Veh. vs PD-1 Ab, \**P* = 0.0124, Veh. vs Lac, \**P* = 0.0318, Veh. vs Combo, \*\**P* = 0.0013 for IFNγ⁺; Veh. vs PD-1 Ab, \*\*\**P* = 0.0005, Veh. vs Lac, \*\*\**P* = 0.0007, Veh. vs Combo, \*\*\*\**P* < 0.0001 for PER⁺. (G): Statistics of the proportion of CD4⁺IFNγ⁺ in tumor of Fig. 7J. Data are representative of three independent experiments and were analyzed by Student's *t* test. Error bars denote mean ± SD. *P* value: Veh. vs PD-1 Ab, \*\**P* = 0.0013, Veh. vs Lac, \*\**P* = 0.0090, Veh. vs Combo, \*\*\**P* = 0.0004 for IFNγ⁺. (H) Representative images of H&E staining of the indicated organs of treated mice. Scale bar: 100 μm. Source data are available online for this figure.

