## [Peer Review File · EMBO Molecular Medicine]

Lanatoside C activates the E3 ligase STUB1 to inhibit FOXP3 transcriptional activity and promote antitumor immunity

Qian Zhou, Tong Yang, Xixi Yu, Bo Li, Jin Liu, Yongxin Mao, Rongxiang Guo, Zhuo Feng, Li Zhou, Guandi Zeng, nan li, Jinxia Liang, lu liu, Pengju feng, Hong-Bing Shu, and Liang Chen

Corresponding authors: Liang Chen (chenliang@jnu.edu.cn) , Hong-Bing Shu (shuh@whu.edu.cn), Qian Zhou (zhouqian@jnu.edu.cn)

Review Timeline:

Submission Date:	6th Aug 24
Editorial Decision:	28th Aug 24
Revision Received:	27th Nov 24
Editorial Decision:	18th Dec 24
Revision Received:	29th Dec 24
Accepted:	3rd Feb 25

Editor: Jingyi Hou

Transaction Report:

28th Aug 2024

Dear Dr. Zhou,

Thank you again for submitting your work to EMBO Molecular Medicine. We have now heard back from the two referees who agreed to evaluate your manuscript. As you will see from the reports below, the referees find your study of potential interest. However, they raise a series of concerns, which should be convincingly addressed in a major revision of the present manuscript.

The referees' recommendations are relatively straightforward, so there is no need for me to reiterate the points listed below. All the issues raised by the reviewers need to be carefully addressed.

We would welcome the submission of a revised version within three months for further consideration. As you may already know, our editorial policy allows in principle a single round of major revision, and it is therefore essential to provide responses to the referees' comments that are as complete as possible.

Please also contact us as soon as possible if similar work is published elsewhere. If other work is published, we may not be able to extend the revision period beyond three months.

I look forward to receiving your revised manuscript.

Kind regards,
Jingyi

Jingyi Hou
Editor
EMBO Molecular Medicine

We require:

- 1) A .docx formatted version of the manuscript text (including legends for main figures, EV figures and tables). Please make sure that the changes are highlighted to be clearly visible.
- 2) Individual production quality figure files as .eps, .tif, .jpg (one file per figure). For guidance, download the 'Figure Guide PDF': (<https://www.embopress.org/page/journal/17574684/authorguide#figureformat>).
- 3) A .docx formatted letter INCLUDING the reviewers' reports and your detailed point-by-point responses to their comments. As part of the EMBO Press transparent editorial process, the point-by-point response is part of the Review Process File (RPF), which will be published alongside your paper.
- 4) A complete author checklist, which you can download from our author guidelines (<https://www.embopress.org/page/journal/17574684/authorguide#submissionofrevisions>). Please insert information in the checklist that is also reflected in the manuscript. The completed author checklist will also be part of the RPF.
- 5) Please note that all corresponding authors are required to supply an ORCID ID for their name upon submission of a revised manuscript.
- 6) It is mandatory to include a 'Data Availability' section after the Materials and Methods. Before submitting your revision, primary

datasets produced in this study need to be deposited in an appropriate public database, and the accession numbers and database listed under 'Data Availability'. Please remember to provide a reviewer password if the datasets are not yet public (see <https://www.embopress.org/page/journal/17574684/authorguide#dataavailability>).

12) Author contributions: You will be asked to provide CRediT (Contributor Role Taxonomy) terms in the submission system. These replace a narrative author contribution section in the manuscript.

13) A Conflict of Interest statement should be provided in the main text.

14) Every published paper now includes a 'Synopsis' to further enhance discoverability. Synopses are displayed on the journal webpage and are freely accessible to all readers. They include a short stand first (maximum of 300 characters, including space) as well as 2-5 one-sentences bullet points that summarizes the paper. Please write the bullet points to summarize the key NEW findings. They should be designed to be complementary to the abstract - i.e. not repeat the same text. We encourage inclusion

of key acronyms and quantitative information (maximum of 30 words / bullet point). Please use the passive voice. Please attach these in a separate file or send them by email, we will incorporate them accordingly.

15) All Materials and Methods need to be described in the main text using our 'Structured Methods' format, which is required for all research articles. According to this format, the Methods section includes a Reagents and Tools Table (listing key reagents, experimental models, software and relevant equipment and including their sources and relevant identifiers) followed by a Methods and Protocols section describing the methods using a step-by-step protocol format.

The Reagents and Tools Table can be downloaded from our author guidelines (<https://www.embopress.org/page/journal/17574684/authorguide#structuredmethods>)

***** Reviewer's comments *****

Referee #1 (Comments on Novelty/Model System for Author):

Treg cells are major obstacle in cancer therapy. In this study, the authors develop an interesting screening technique and found that Lanatoside C, an FDA-approved drug, could inhibit FOXP3 transcriptional activity and promote antitumor immunity, thus has potential translational values.

Referee #1 (Remarks for Author):

The potential interesting manuscript by Dr. Zhou and colleagues investigated the potential inhibitors of FOXP3 transcriptional activity by screening FDA-approved drugs, and identified Lanatoside C (Lac) as an effective disruptor of the FOXP3-RUNX1 complex. Functional assays revealed that Lac attenuated the differentiation and proliferation of Tregs as well as their inhibitory activities on CTLs. Furthermore, the authors explored the mechanisms through which Lac induces the degradation of RUNX1. Notably, they demonstrated the anti-tumor efficacy of Lac in two mouse models of lung cancer and highlighted its potential synergistic effect with ICB in delaying tumor growth. Overall, this study is well designed and the results obtained are interesting and convincing. The immunosuppressive function of Tregs is a major challenge in cancer therapy. Considering Lac is FDA-approved, its discovery as a FOXP3 inhibitor offers a unique advantage for expedited clinical application, and thus has potential translational values. I have only a few minor concerns.

1. In Figure 7, Lac dramatically enhanced PD-1 antibody to shrink tumors. However, their impact on reduced Tregs infiltration and enhanced expression of IFN γ and perforin (as shown with the flow cytometry data in Figure 7 and Figure S9) seems relatively mild. One possible reason could be due to the method used, since flow cytometry analysis might not truly represent the immune-microenvironment in situ tumors. The authors are encouraged to include the image data from Immunohistochemistry or IF staining on these samples.
2. The gating strategy and analysis in flow cytometry data could be optimized to get better separation on cell populations. Also, ELISA could be employed to determine the cytokine production in some experiments.
3. The dosage of Lanatoside C: High dose of Lanatoside C is toxic for patients in clinic. The authors should discuss whether the concentration of Lanatoside C used in this study could be related to clinic.
4. For clarity, the names of fluorescence dye in the flow data should be omitted.
5. The screen platform is interesting and useful. Could this technique be used for other model? The authors could add their view in the discussion.

Referee #2 (Comments on Novelty/Model System for Author):

The study's core finding that an FDA-approved drug, Lanatoside C, can inhibit Treg function and enhance anti-tumor immunity is quite exciting. However, several technical issues, especially the poor Foxp3 FACS staining, undermined the quality of the study. Most of the quality issues are fixable with a proper revision.

Referee #2 (Remarks for Author):

In this manuscript, Zhou and colleagues identified Lanatoside C (Lac) as an inhibitor of Foxp3 transcriptional activity through a high-throughput screen of FDA-approved drugs. They showed that Lac accelerates the degradation of RUNX1 protein, a partner associated with Foxp3 for its function, via binding to E3 ligase STUB1 and promoting RUNX1 polyubiquitination. Finally, Lac treatment alone or in combination with PD-1 inhibitor promotes anti-tumor immunity in mouse lung cancer models due to

reduced Treg activity. The main strength of the study is the identification of a new therapeutic potential for the FDA-approved drug Lanatoside C in promoting anti-tumor immunity. Some weaknesses, mostly involving Treg characterization, are also identified as listed below.

1. The most prominent issue throughout the data presented in the manuscript is the poor quality of Foxp3 staining in FACS. Instead of a clear Foxp3+ population separated from a Foxp3- population, the Foxp3 staining in all the figures only showed a small shift. It is not clear how the Foxp3+ gates were drawn since there is no separation of the positive and negative populations. This issue compromised the confidence in the key conclusions of the study. Foxp3 FACS staining is a routine procedure that has been conducted in numerous labs. I suggest the authors use the FJK-16s clone from eBioscience with their Foxp3 staining buffer set to achieve proper Foxp3 staining. It's understandable that not all Foxp3 stains need to be repeated, but some of the key in vitro assays should be redone with proper Foxp3 staining to enhance the quality of the overall study.
2. The authors showed very nice data on how Lac binds to STUB1's TPR domain. However, it is not clear how the interaction between Lac and STUB1 can enhance the ubiquitination and degradation of RUNX1.
3. RUNX1 is important for the development of multiple lymphoid and myeloid cell lineages. For example, thymic T cell development is disrupted in RUNX1 knockout mice. Mutations in RUNX1 could lead to leukemia. The current study has not adequately addressed the safety issue of Lac treatment. At a minimum, the authors need to analyze the composition of the thymic and spleen T cell subsets after two weeks of Lac treatment.
4. In Fig. 1C, Foxp3 mRNA should be quantified along with CD25, CTLA4, and IFN γ .
5. For CFSE measurement of cell division in Fig. 1I and Fig. 2A, all divided cell populations should be included (all peaks except the one on the most right side) instead of cells with the most divisions (the peak on the most left side). Cell divisions can also be quantified using the division index and proliferation index calculated by the Flowjo software.
6. Please add quantification to the Western-blot results in Fig. 3H and Fig. 4I. The differences in band intensity are not evident in these experiments.
7. The STUB1 co-IP bands look poor in Fig. 4D. Also, Lac treatment in this experiment did not degrade RUNX1 more compared to DMSO treatment. Is there an explanation?
8. Please show statistics on the FACS data in Fig. 7G-7J since the differences in these experiments were not evident in some cases.

***** Reviewer's comments *****

Referee #1 (Comments on Novelty/Model System for Author):

Treg cells are major obstacle in cancer therapy. In this study, the authors develop an interesting screening technique and found that Lanatoside C, an FDA-approved drug, could inhibit FOXP3 transcriptional activity and promote antitumor immunity, thus has potential translational values.

Referee #1 (Remarks for Author):

The potential interesting manuscript by Dr. Zhou and colleagues investigated the potential inhibitors of FOXP3 transcriptional activity by screening FDA-approved drugs, and identified Lanatoside C (Lac) as an effective disruptor of the FOXP3-RUNX1 complex. Functional assays revealed that Lac attenuated the differentiation and proliferation of Tregs as well as their inhibitory activities on CTLs. Furthermore, the authors explored the mechanisms through which Lac induces the degradation of RUNX1. Notably, they demonstrated the anti-tumor efficacy of Lac in two mouse models of lung cancer and highlighted its potential synergistic effect with ICB in delaying tumor growth. Overall, this study is well designed and the results obtained are interesting and convincing. The immunosuppressive function of Tregs is a major challenge in cancer therapy. Considering Lac is FDA-approved, its discovery as a FOXP3 inhibitor offers a unique advantage for expedited clinical application, and thus has potential translational values. I have only a few minor concerns.

Reply: We thank Reviewer for finding our work interesting and convincing.

1. In Figure 7, Lac dramatically enhanced PD-1 antibody to shrink tumors. However, their impact on reduced Tregs infiltration and enhanced expression of IFN γ and perforin (as shown with the flow cytometry data in Figure 7 and Figure S9) seems relatively mild. One possible reason could be due to the method used, since flow cytometry analysis might not truly represent the immune-microenvironment in situ tumors. The authors are encouraged to include the image data from Immunohistochemistry or IF staining on these samples.

Reply: We thank Reviewer for these great suggestions! Following Reviewer's suggestion, we have checked Tregs infiltration (CD4⁺FOXP3⁺) and expression of IFN γ (CD8⁺IFN γ ⁺) and perforin (CD8⁺ perforin⁺) through IF staining on these

samples. Our results argued that Lac drastically reduced the proportion of CD4⁺FOXP3⁺Tregs in immune-microenvironment in situ tumors (updated in new Fig. EV5B & EV5C). Unfortunately, Perforin and IFN γ staining didn't work well IF (Fig.1 to Reviewer). We agree with Reviewer that FACS by itself may underestimate the immune reaction in tumor in situ. However, due to limitation of reagents, we are unable to present data from IF.

Fig.1 to Reviewer

A & B: Immunofluorescence analysis of expression of IFN γ and perforin by CD8⁺T cells. KRAS^{G12D} bearing lung cancer mouse received indicated treatments for 2 weeks. Part of lung tissues were fixed for Immunofluorescence analysis expression of IFN γ (A) and perforin (B) by CD8⁺T cells. Scale bar: 100 μ m.

2. The gating strategy and analysis in flow cytometry data could be optimized to get better separation on cell populations. Also, ELISA could be employed to determine the cytokine production in some experiments.

Reply: Thanks so much for your suggestions. The gating strategy and analysis in flow cytometry data are optimized. A new FOXP3 antibody (FJK-16s clone from eBioscience) and its Foxp3 staining buffer (00-5523-00, eBioscience) were also employed to repeated some key in vitro assay to get better separation on cell

populations (New Fig. 1D-G & Fig. EV1B & D& F& H & Fig.2 to Reviewer).

Fig.2 to Reviewer

A-C: Gating strategy for analyzing the expression of Perforin, IL10, CD25 or CTLA4 from mouse spleen or lymph node. New FOXP3 antibody (FJK-16s clone from eBioscience) and Foxp3 staining buffer (00-5523-00, eBioscience) were used to stain FOXP3 according to the manufacturer's manual.

Following Reviewer's suggestion, ELISA was employed to determine expression of IL10 secreted by Treg cells. Our results suggested that Lac. significantly suppress IL10 expression by Tregs (New Fig. 1H & Fig.3 to reviewer).

Fig.3 to Reviewer

A: Mouse IL10 ELISA standard curve; B: ELISA analysis of the impact of Lac on IL10

expression by Tregs. Tregs (FOXP3⁺CD25⁺) cells were treated without or with Lac for 48 or 72 hours before ELISA analysis.

3. The dosage of Lanatoside C: High dose of Lanatoside C is toxic for patients in clinic. The authors should discuss whether the concentration of Lanatoside C used in this study could be related to clinic.

Reply: We agree that Lanatoside C dose in our study is rather high (6mg/kg every two days). Our current paper is a proof of principle. If lucky enough, our work was supported for clinical translation, dosage will be carefully titrated in Stage I trial. Moreover, Lac could be administered into tumor directly or through help from nano-materials. We have discussed this issue in page 20 (line 445-line 452).

4. For clarity, the names of fluorescence dye in the flow data should be omitted.

Reply: Thanks so much for these suggestions. We have removed the names of fluorescence dye in the flow data.

5. The screen platform is interesting and useful. Could this technique be used for other model? The authors could add their view in the discussion.

Reply: We thank Reviewer for offering us the opportunity to highlight the potency and convenience of our screen platform. Actually, we have successfully applied similar platform in measuring the inhibitory function of small molecular chemicals on interaction between PD-1 and SHP2 (EMBO Molecular Medicine. 2020;12(6):e11571).

We have now discussed the potency and applicability of this platform in Discussion part (page17, line 384-line 394).

Referee #2 (Comments on Novelty/Model System for Author):

The study's core finding that an FDA-approved drug, Lanatoside C, can inhibit Treg function and enhance anti-tumor immunity is quite exciting. However, several technical issues, especially the poor Foxp3 FACS staining, undermined the quality of the study. Most of the quality issues are fixable with a proper revision.

Reply: We thank Reviewer for finding our study exciting.

Referee #2 (Remarks for Author):

In this manuscript, Zhou and colleagues identified Lanatoside C (Lac) as an inhibitor of Foxp3 transcriptional activity through a high-throughput screen of FDA-approved drugs. They showed that Lac accelerates the degradation of RUNX1 protein, a partner associated with Foxp3 for its function, via binding to E3 ligase STUB1 and promoting RUNX1 polyubiquitination. Finally, Lac treatment alone or in combination with PD-1 inhibitor promotes anti-tumor immunity in mouse lung cancer models due to reduced Treg activity. The main strength of the study is the identification of a new therapeutic potential for the FDA-approved drug Lanatoside C in promoting anti-tumor immunity. Some weaknesses, mostly involving Treg characterization, are also identified as listed below.

1. The most prominent issue throughout the data presented in the manuscript is the poor quality of Foxp3 staining in FACS. Instead of a clear Foxp3⁺ population separated from a Foxp3⁻ population, the Foxp3 staining in all the figures only showed a small shift. It is not clear how the Foxp3⁺ gates were drawn since there is no separation of the positive and negative populations. This issue compromised the confidence in the key conclusions of the study. Foxp3 FACS staining is a routine procedure that has been conducted in numerous labs. I suggest the authors use the FJK-16s clone from eBioscience with their Foxp3 staining buffer set to achieve proper Foxp3 staining. It's

understandable that not all Foxp3 stains need to be repeated, but some of the key in vitro assays should be redone with proper Foxp3 staining to enhance the quality of the overall study.

Reply: Thanks so much for these great suggestions! A new FOXP3 antibody (FJK-16s clone from eBioscience) and its Foxp3 staining buffer (00-5523-00, eBioscience) were employed to repeat key in vitro assays. The new system greatly improved the FOXP3 staining (New Fig. 1D-G & Fig. EV1B & D & F& H) and better FOXP3⁺ population resolution (Fig.2 to Reviewer). Our current data clearly showed that Lac treatment inhibited CD25, CTLA4, Perforin (PER) and IL10 expression in Tregs.

Fig.2 to Reviewer

A-C: Gating strategy for analyzing the expression of Perforin, IL10, CD25 or CTLA4 from mouse spleen or lymph node. New FOXP3 antibody (FJK-16s clone from eBioscience) and Foxp3 staining buffer (00-5523-00, eBioscience) were used to stain FOXP3 according to the manufacturer's manual.

2. The authors showed very nice data on how Lac binds to STUB1's TPR domain.

However, it is not clear how the interaction between Lac and STUB1 can enhance the ubiquitination and degradation of RUNX1.

Reply: Our data (Fig. 4D) showed that Lac treatment enhanced ability of STUB1 to bind RUNX. Please refer to lane 1 and lane 2 in the experiment: MG132 stabilized RUNX1; In this case, Lac enhanced the ability of RUNX1 to immunoprecipitate STUB1.

We believe that binding of Lac to STUB1's TPR domain altered the structure of STUB1, such that the interaction between STUB1 and RUNX1 is enhanced. We tried to run a molecular simulation on that. However, due to our limited capacity on the simulation, we fail in these attempts. Therefore, we are unable to show the detailed structural information on this point. We hope that Reviewer is persuaded by our data that Lac binds STUB1 to form a STUB1/Lac complex and that this STUB1 complex has enhanced binding ability to RUNX1 for ubiquitylation.

3. RUNX1 is important for the development of multiple lymphoid and myeloid cell lineages. For example, thymic T cell development is disrupted in RUNX1 knockout mice. Mutations in RUNX1 could lead to leukemia. The current study has not adequately addressed the safety issue of Lac treatment. At a minimum, the authors need to analyze the composition of the thymic and spleen T cell subsets after two weeks of Lac treatment.

Reply: Thanks so much for pointing this important point to us. Following Reviewer's suggestion, we checked the composition of T cell subsets in thymic and spleen after

two weeks of Lac treatment.

We found that the spleens were larger in LAC treatment group compared to the Vehicle group (Fig. 3 to Reviewer A). We saw a decrease in proportion of CD4⁺ and CD8⁺ cells in spleens. However, the ratio of CD8⁺ to CD4⁺ cells remain unchanged (Fig. 3 to Reviewer B & C). Importantly, we saw that LAC treatment significantly reduced the proportion of TREG (FOXP⁺CD25⁺). We also noticed a decrease in TH1 (CD4⁺IFN γ ⁺) and TH2 (CD4⁺IL4⁺) populations in spleens. In contrast, percentage of TH17 (CD4⁺IL17A⁺) cells remains constant (Fig. 3 to Reviewer D-G) in spleens.

We also checked the alteration of T cell subsets in thymus. We found that the percentage of double positive (CD4⁺CD8⁺) thymocytes were reduced. In contrast, CD8 single positive (CD8⁺) thymocytes were increased and CD4 single positive (CD4⁺) thymocytes had no remarkable change (Fig. 4 to Reviewer H & I). Taken together, our data showed that 2 weeks treatment with LAC reduced Treg percentage while have limited impact on overall immune components.

Fig. 4 to Reviewer

Mice (C57BL/6J) were randomized for treatment (n=4) with PBS or Lac for two weeks.

A: Spleens were dissected to photograph; B: FACS analysis of proportion of CD4⁺ and CD8⁺ cells in spleen after 2 weeks of Lac treatment; C: Statistics of Fig. 3 to Reviewer B; D-G: FACS analysis of proportion of TREG (FOXP3⁺CD25⁺), TH1 (CD4⁺IFNγ⁺), TH2 (CD4⁺IL4⁺) and TH17 (CD4⁺IL17A⁺) cell populations in the spleen after 2 weeks of Lac treatment (D & F). Statistics of Fig. 3 to Reviewer D & F (E & G); H: FACS analysis of proportion of CD4⁺ SP, CD8⁺ SP and CD4⁺CD8⁺ DP cells in thymus after 2 weeks of Lac treatment; I: Statistics of H.

4. In Fig. 1C, Foxp3 mRNA should be quantified along with CD25, CTLA4, and IFNγ.

Reply: Thanks so much for this great suggestion. We conducted qRT-PCR to quantify

Foxp3 mRNA along with *Cd25*, *Ctla4*, *Il10* and *Ifn γ* . We found that Lac potentially inhibited the transcription of *Foxp33*, *Il10*, *Cd25* and *Ctla4*, but not *Ifn γ* (new Fig. 1C).

5. For CFSE measurement of cell division in Fig. 1I and Fig. 2A, all divided cell populations should be included (all peaks except the one on the most right side) instead of cells with the most divisions (the peak on the most left side). Cell divisions can also be quantified using the division index and proliferation index calculated by the Flowjo software.

Reply: We thank Reviewer for pointing out this important issue to us. Follow Reviewer's suggestion, Division index (DI) and proliferation index (PI) were calculated by the FlowJo software. Results were updated in New Fig. 1I & EV1L & 2A & EV2A.

6. Please add quantification to the Western-blot results in Fig. 3H and Fig. 4I. The differences in band intensity are not evident in these experiments.

Reply: Thanks so much for bringing this point to our attention. We quantified the bands against β -actin. The results are updated in New Appendix Fig. S3G (Fig. 3H) & Fig. 4J (Fig. 4I). Our results indicated that Lac treatment shortened the half-life of both ectopically expressed and endogenous RUNX1.

7. The STUB1 co-IP bands look poor in Fig. 4D. Also, Lac treatment in this experiment

did not degrade RUNX1 more compared to DMSO treatment. Is there an explanation?

Reply: We thank Reviewer for pointing this out to us. In order to clarify the degradation of RUNX1, we extended the treatment time of Lac and repeated this experiment (New Fig. 4D). Our results clearly showed that Lac promoted the association between RUNX1 and STUB1.

8. Please show statistics on the FACS data in Fig. 7G-7J since the differences in these experiments were not evident in some cases.

Reply: Thanks so much for this great suggestion. The statistics on the FACS data of Fig. 7G-7J were included in New Fig. EV5A & EV5F & EV5G.

18th Dec 2024

Dear Liang,

Thank you for submitting your revised manuscript to EMBO Molecular Medicine. We have now received the enclosed report from the two referees who re-assessed your work. As you will see, the referees are now supportive, and I am pleased to inform you that we will be able to accept your manuscript pending the following amendments:

1. Please remove the Authors' Contribution section from the manuscript file.
2. The current synopsis image is too large. Please provide the figure as a PNG file 550 px wide x 300-600 px high.
3. Appendix needs to be uploaded as a PDF file.
4. ORCID IDs should be removed from page 33.
5. Methods
 - "Materials and Methods" should be renamed to "Methods".
 - Remove redundant information already included in the Reagents and Tools table, such as Reagents, Experimental instruments and equipment.
 - Please rewrite the "Plasmid constructs" section to include more details. Remove the related information in Reagents and Tools table.
 - Ethics statements should be removed from page 32 and integrated into the relevant sections on animal experiments in the "Methods" section.
 - For animal work, confirm that all experiments were performed in accordance with relevant guidelines and regulations. Gender, age and genetic background must be indicated, along with housing conditions.
6. "Conflict of interest" should be renamed to "Disclosure Statement and Competing Interests" and placed after "Acknowledgments".
7. "The paper explained" is too brief. Please expand this section and incorporate it into the manuscript file. You may refer to any of our published articles for guidance on the appropriate length and structure.
8. The Synopsis and Bullet points are also too concise. Please refer to any of our published article for an example.
9. Data availability:
 - Before submitting your revision, the mass spectrometry datasets produced in this study need to be deposited in an appropriate public database, and the accession numbers and database listed under 'Data Availability'.
 - Please update the Data availability information in the Author checklist accordingly.
10. EV tables and datasets
 - The current Tables EV1, 3, and 4 are quite large. Please update them to EV Datasets and upload them as Excel tables. Include the legends within the Dataset in a separate sheet labelled 'Legend'. Also, update their callouts (in the manuscript file and Reagents and Tools table) to Dataset EV#.
 - Please update the callouts of the remaining EV tables accordingly.
 - There are Chinese characters in the current Table EV3 - please correct this.
11. Funding: the text should be removed from the Comments box, and each funder and its grant number should be provided as separate entries (More Funders option): Guangdong basic and applied basic research foundation(2024A1515030238), Guangzhou Science and Technology program City-University Joint Funding Project (2024A03J0596), Open Project Funded by the MOE Key Laboratory of Tumor Molecular Biology (202302)
12. Source data:
 - Source data for EV and Appendix figures need to be combined into one zip folder.
 - For main figures: all SD need to be grouped into zip folders and uploaded as one folder per figure, using the nomenclature "manuscriptID_SourceDataForFigure x"
 - Please provide numerical data as individual .xls (including a tab describing the data) or as .csv files (including a separate README file) instead of .PRISM, .pzfx, or .fcs file.
13. Please address the following issues in figure legends:
 - Please note that the legend for figure 1J is missing in the manuscript. This needs to be rectified.
 - Please note that the exact p values are not provided in the legends of figures 1C, H, J; 2B, D, F, H; 3E, 4F, J; 5B, D, G, L;

6B,C, E, G, I, K, M; 7B, D, F; EV1 C, E, G, I, K, M; EV2F; EV3 B, F; EV4G; EV5 A, C, E, F; Supplementary figure(s) 1G; 2B, 3F, G;

- Please note that in figure EV1 C, E, G, I, K, M, O; EV3 B, F; EV4G; EV5A, C, E, F; there is a mismatch between the annotated p-values in the figure legend and the annotated p values in the figure file that should be corrected.

- Please indicate what */ **/ ***/ **** represents; if this represents p value(s), please indicate the statistical test used and where appropriate and the exact p value in the legend(s) of supplementary figure(s) 5B, D, F. "

- Please note that information related to n is missing in the legends of figures supplementary figures 5B, D, F.

- Please note that the error bars are not defined in the legends of supplementary figures 5B, D, F

- Please note that the measure of center for the error bars needs to be defined in the legends of figures 1C, H, J; 2B, C, F, H; 3E, 4F,J; 5B, D, G, L; 6B, C, E, G, I, K, M; 7B, D, F; EV1 C, E, G, I, M, K, O; EV2 B, D, F, H; 3B, F; 4G, 5A, C, E, F, G; Supplementary figure(s) 1A, B, F, G; 2B, D; 3C, D, E, F, G; 4A, C, D, F.

Please feel free to let me know if you have any questions. I look forward to seeing a revised form of your manuscript soon.

Kind regards,
Jingyi

Jingyi Hou
Editor
EMBO Molecular Medicine

*** Instructions to submit your revised manuscript ***

To submit your manuscript, please follow this link:

<https://embomolmed.msubmit.net/cgi-bin/main.plex>

***** Reviewer's comments *****

Referee #1 (Comments on Novelty/Model System for Author):

This study is well designed and the results obtained are interesting. The immunosuppressive function of Tregs is a major challenge in cancer therapy. Considering Lac is FDA-approved, its discovery as a FOXP3 inhibitor offers a unique advantage for expedited clinical application, and thus has potential translational values.

Referee #1 (Remarks for Author):

The authors addressed most of the concerns raised by the Reviewers and have revised the manuscript accordingly.

Referee #2 (Remarks for Author):

The revised manuscript has fully addressed the reviewer's concerns. Congratulations to the authors for their excellent work!

The authors addressed the remaining editorial issues.

3rd Feb 2025

Dear Liang,

Happy Chinese New Year! I am pleased to inform you that your manuscript is accepted for publication and is now being sent to our publisher to be included in the next available issue of EMBO Molecular Medicine.

If you have any questions, please do not hesitate to contact the Editorial Office.

Thank you for sending this interesting work to EMBO Molecular Medicine!

Kind regards,
Jingyi

Jingyi Hou
Senior Editor
EMBO Molecular Medicine
